# SpQR: A Sparse-Quantized Representation for Near-Lossless LLM Weight Compression

**Tim Dettmers**[*†1]    **Ruslan Svirschevski**[*2,3]    **Vage Egiazarian**[*2,3]    **Denis Kuznedelev**[*3,4]
**Elias Frantar**[5]    **Saleh Ashkboos**[6]    **Alexander Borzunov**[2,3]    **Torsten Hoefler**[6]    **Dan Alistarh**[5,7]
[1] University of Washington    [2] HSE University    [3] Yandex    [4] Skoltech
[5] IST Austria    [6] ETH Zurich    [7] NeuralMagic

## Abstract

Recent advances in large language model (LLM) pretraining have led to high-quality LLMs with impressive abilities. By compressing such LLMs via quantization to 3-4 bits per parameter, they can fit into memory-limited devices such as laptops and mobile phones, enabling personalized use. Quantizing models to 3-4 bits per parameter can lead to moderate to high accuracy losses, especially for smaller models (1-10B parameters), which are suitable for edge deployment. To address this accuracy issue, we introduce the Sparse-Quantized Representation (SpQR), a new compressed format and quantization technique that enables for the first time *near-lossless* compression of LLMs across model scales while reaching similar compression levels to previous methods. SpQR works by identifying and isolating *outlier weights*, which cause particularly large quantization errors, and storing them in higher precision while compressing all other weights to 3-4 bits, and achieves relative accuracy losses of less than $1\%$ in perplexity for highly-accurate LLaMA and Falcon LLMs. This makes it possible to run a 33B parameter LLM on a single 24 GB consumer GPU without performance degradation at 15% speedup, thus making powerful LLMs available to consumers without any downsides. SpQR comes with efficient algorithms for both encoding weights into its format, as well as decoding them efficiently at runtime. Specifically, we provide an efficient GPU inference algorithm for SpQR, which yields faster inference than 16-bit baselines at similar accuracy while enabling memory compression gains of more than 4x.

## 1 Introduction

Pretrained large language models (LLMs) improved rapidly from task-specific performance (Wang et al., 2018; Devlin et al., 2019; Radford et al., 2019), to performing well on general tasks if prompted with instructions (Brown et al., 2020; Wei et al., 2021; OpenAI, 2023). While the improved performance can be attributed to scaling in training data and parameters (Kaplan et al., 2020; Chowdhery et al., 2022) recent trends focused on smaller models trained on more data, that are easier to use at inference time (Hoffmann et al., 2022; Biderman et al., 2023; Touvron et al., 2023). For example, the 7B parameter LLaMA model trained on 1 trillion tokens achieved an average performance only slightly lower than GPT-3 (Brown et al., 2020) despite being 25x smaller. Current techniques for LLM compression can shrink these models further by a factor of about 4x, while preserving their performance (Dettmers et al., 2022; Xiao et al., 2022; Frantar et al., 2022a; Dettmers & Zettlemoyer, 2022). This yields performance levels comparable to the largest GPT-3 model, with major reductions in terms of memory requirements. With such improvements, well-performing models could be efficiently served on end-user devices, such as laptops.

The main challenge is to compress models enough to fit into such devices while also preserving generative quality. Specifically, studies show that, although accurate, existing techniques for 3 to 4-bit quantization still lead to significant accuracy degradation (Dettmers & Zettlemoyer, 2022; Frantar et al., 2022a). Since LLM generation is sequential, depending on previously-generated tokens, small errors can accumulate and lead to severely diverging outputs. To ensure reliable quality, it is critical to design quantization that does not degrade predictive performance compared to the 16-bit model.

---

[*]Equal contribution.
[†]Corresponding author: `dettmers@cs.washington.edu`

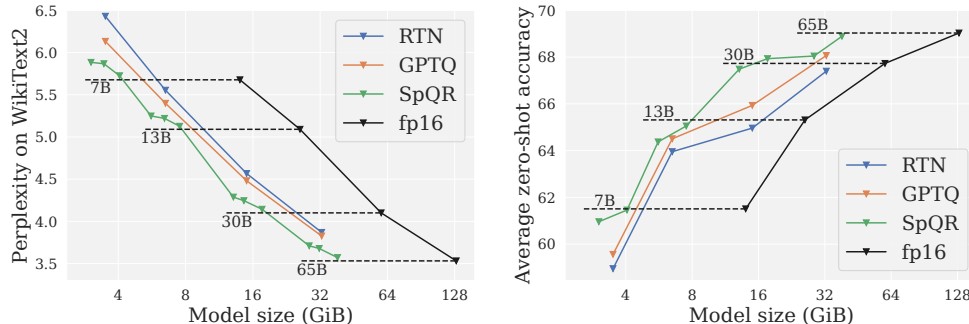

Figure 1: Compressed LLM performance for LLaMA models. (**left**) LM loss on WikiText2 vs model size. (**right**) Average performance on zero-shot tasks vs model size.

In this work, we introduce Sparse-Quantized Representations (SpQR), a hybrid sparse-quantized format which can be used to compress accurate LLMs to 3-4 bits per parameter while staying *near-lossless*: specifically, SpQR is the first weight quantization method which is able to reach high compression ratios while inducing end-to-end error of less than 1% relative to the dense baseline. This 1% relative threshold follows the MLCommons standards (Reddi et al., 2020); since LLM accuracy is usually measured in the stringent *perplexity* metric, this is very challenging to achieve.

Our compression approach is motivated by a new analysis showing that LLM weight quantization errors exhibit both vertical and horizontal group correlations, corresponding to systematic large errors corresponding to input and output dimensions. While outlier input features have been observed before (Dettmers et al., 2022; Xiao et al., 2022), our work is the first to demonstrate that similar outliers occur *in the weights, for particular output hidden dimensions*. Unlike input feature outliers, the output hidden dimension outliers occur only in small segments for a particular output hidden dimension.

In SpQR, we resolve the difficulties posed by these correlations by combining two ideas. First, we isolate *outlier weights*, whose quantization induces disproportionately high errors: these weights are kept in high precision, while the other weights are stored in lower bitwidth. Second, we implement a variant of grouped quantization with very small group size, e.g., 16 contiguous elements, but we show that one can quantize the quantization scales themselves to a 3-bit representation.

To convert a given pretrained LLM into SpQR format, we extend the post-training quantization (PTQ) approach recently introduced by GPTQ (Frantar et al., 2022a). Specifically, this method passes calibration data through the uncompressed model; to compress each layer, it applies a layer-wise solver with respect to the L2 error between the outputs of the uncompressed model, and those of the quantized weights. Our technique splits this process into two steps: an "outlier detection" step, in which we isolate weights whose direct quantization has outsize impact on layer output behavior, and a compression step, in which $\geq 99\%$ of weights are compressed to low-bitwidth. Furthermore, the whole representation is rendered more efficient by compressing the quantization metadata. Our quantization algorithm isolates such outliers and efficiently encodes a given model in SpQR format.

One key challenge in exploiting the SpQR format is computational efficiency, as sparse representations generally have poor GPU support. We circumvent this issue by developing a sparse-matrix multiplication algorithm that is specialized to our format. To use SpQR for generative inference, we combine sparse-matrix multiplication for the outliers with dense-quantized matrix multiplication for the 3-4 bit "base" weights. Our experimental evaluation, across LLMs from the LLaMA (Touvron et al., 2023), Falcon (TII UAE, 2023a), and OPT (Zhang et al., 2022) families, shows that SpQR provides state-of-the-art trade-offs in terms of accuracy versus model size (see Figure 1 for an illustration). For instance, SpQR can provide 3.4x compression without any degradation in perplexity, while also being 20-30% faster for LLM generation compared to 16-bit inference.

## 2 RELATED WORK

We focus our discussion on related *post-training quantization (PTQ) methods* (Nagel et al., 2020), referring the reader to the recent survey of Gholami et al. (2021) for full background on quantization. PTQ is popular for *one-shot compression* of models with various sizes, based on a limited amount of calibration data, using accurate solvers. Most PTQ methods, such as AdaRound (Nagel et al., 2020), BitSplit (Wang et al., 2020), AdaQuant (Hubara et al., 2021), BRECQ (Li et al., 2021), or

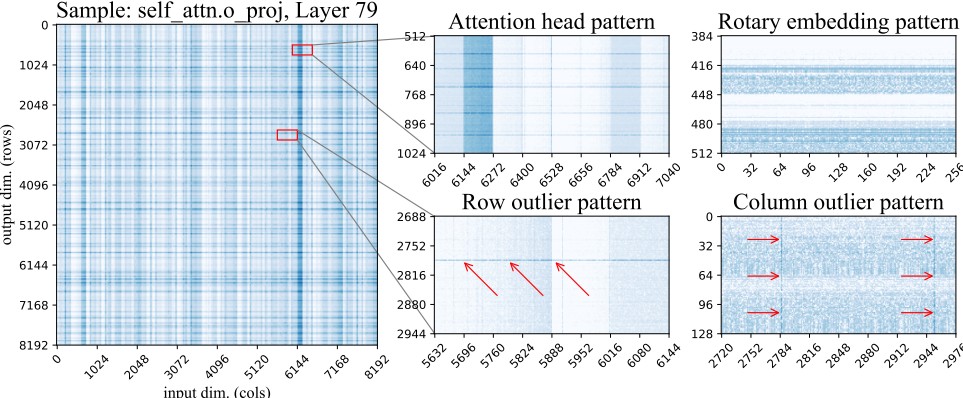

Figure 2: Weight log-sensitivities from the last attention layer of LLaMA-65B. Dark-blue shades indicate higher sensitivity. The image on the left is a high-level view, resized to 1:32 scale with max-pooling. The two images in the middle are zoomed in from the main figure. The two images are zoomed in sensitivities of weights from to other weight matrices.

OBQ (Frantar et al., 2022b) were designed for vision or small-scale language models. These recent approaches tend to use accurate solvers, which would not scale to LLMs in terms of computational or memory cost, as they are 10-1000x larger in size.

Recently, there has been significant interest in obtaining accurate post-training methods that scale to such massive models. Due to computational constraints, early work such as ZeroQuant (Yao et al., 2022), LLM.int8() (Dettmers et al., 2022), and nuQmm (Park et al., 2022) used direct rounding of weights to the nearest quantization level, while customizing the quantization granularity (i.e., group size) to trade off space for increased accuracy. LLM.int8() (Dettmers et al., 2022) suggested isolating "outlier features" which would be quantized separately to higher bit-width. These approaches are able to induce relatively low error, e.g., 5.5% relative LM Loss increase for LLaMA-7B at 4-bit weight quantization, provided that the quantization granularity is low enough. GPTQ (Frantar et al., 2022a) proposed a higher-accuracy approach (e.g., 4% LM Loss increase in the above setting), which works via an approximate large-scale solver for the problem of minimizing the layer-wise squared error.

Dettmers & Zettlemoyer (2022) provided an in-depth overview of the accuracy-compression trade-offs underlying these methods, establishing that 4-bit quantization is an optimal point for round-to-nearest-based methods, whereas higher compression can be achieved via data-aware methods such as GPTQ. SparseGPT (Frantar & Alistarh, 2023) presented an approach to jointly sparsify LLM weights to medium sparsities, together with quantization of the remaining weights to a fixed given bit-width. One common drawback of existing methods is that the accuracy loss relative to the original model is still significant (see Section 5). This is especially relevant to relatively small but easily deployable models, e.g., in the 7-13B parameter range, where existing methods show drastic accuracy drops. We investigate this question here, and provide a new compression format which can lead to near-lossless 3-4 bits compression in this regime.

A related question is performing both activation and weight quantization. Dettmers et al. (2022); Xiao et al. (2022); Yao et al. (2022) showed that both activations and weights can be quantized to 8-bits.These investigations yield insights into the sources of quantization error, in particular, the former two observe "outlier features" with significantly higher magnitudes in the input/output of large LLMs, which induce higher quantization error, and propose different mitigation strategies.

We analyze this phenomenon from the point of view of weight quantization that goes beyond outlier features in the hidden state. While we find that input feature outliers of the current layer are correlated to hidden unit outliers weight in the previous layer there is not a strict correspondence. Such partially-structured outlier patterns necessitate a fine-grained hybrid compression format that goes beyond algorithms that exploit the column structure of outlier features found in previous work.

We discuss additional LLM quantization work and parallel work that was published as arXiv preprint at the same time as our work in Appendix A.

# 3 QUANTIZATION SENSITIVITY OF LLM WEIGHTS

## 3.1 PARAMETER SENSITIVITY UNDER QUANTIZATION

Not all parameters in a neural network are equally important. Intuitively, a weight could be seen as sensitive to quantization if its rounding error is large, i.e., it is not close to a quantization point, and/or the inputs it is usually multiplied with are large, amplifying even a small rounding error. These simple notions of sensitivity however disregard the fact that LLMs operate on very large vectors with significant correlations: a weight $w_a$ may have a large rounding error while being strongly correlated to another weight $w_b$, meaning that the error of rounding up $w_a$ can be well compensated by rounding down $w_b$. This idea is exploited by modern quantization algorithms (Frantar et al., 2022a; Yao et al., 2022) and can lead to major improvements over vanilla rounding, especially a low bitwidths. Properly capturing this aspect of sensitivity requires a more robust definition.

For computational tractability, we assess sensitivity on a per-layer level using a small set of *calibration inputs* $X$, collected by running them through the model up to the particular layer. We define the sensitivity $s_{ij}$ of some weight $w_{ij}$ in the layer's weight matrix $W$ as the minimum squared difference between the original predictions on $X$ and those of any weight matrix $W'$ where this weight is quantized, i.e., $w'_{ij} = \text{quant}(w_{ij})$: $s_{ij} = \min_{W'} ||WX - W'X||_2^2$ s.t. $w'_{ij} = \text{quant}(w_{ij})$

Crucially, all weights of $W'$ except for $w'_{ij}$ may take on arbitrary, not necessarily quantized, values in order to compensate for the quantization error incurred by rounding $w_{ij}$, thus capturing the correlation aspect discussed above. Further, as we allow continuous values, this problem admits a closed-form solution: $s_{ij} = \frac{(w_{ij} - \text{quant}(w_{ij}))^2}{2(XX^\top)^{-1}}$, as per the Optimal Brain Surgeon framework (Frantar et al., 2022b).

This saliency measure can be approximated efficiently by quantization solvers, such as GPTQ (Frantar et al., 2022a). In more detail, GPTQ quantizes weight matrices column-by-column while in each step adjusting the not-yet-quantized part to compensate for the quantization error in a similar sense as defined above. Consequentially, instead of statically deciding all sensitivities in advance, they can be computed dynamically as the algorithm processes each column, by using the inverse of the Hessian subselection corresponding to all not yet quantized weights. This matrix is already efficiently computed by GPTQ and thus does not impose any additional overheads. The main advantage of this approach is that $s_{ij}$ is always determined based on the most current value of $w_{ij}$ and thus accounts for adjustments due to previously quantized weights as well.

## 3.2 EXPLORING PARAMETER SENSITIVITY

Before we define out main method, SpQR, we provide a motivating analysis of parameter sensitivity which uncovers that the location of sensitive weights in the weight matrix are not random but have particular structures. To highlight these structural elements during the quantization process, we calculate the the per-weight sensitivities and visualize them for the popular and highly-accurate LLaMA-65B model (Touvron et al., 2023). As the quantization method, we use GPTQ quantization to 3-bit, without weight grouping, following (Frantar et al., 2022a). We use C4 (Raffel et al., 2020) as the calibration dataset, and we estimate the error on 128 sequences of 2048 tokens each. Figure 2 depicts the output projection of the last self-attention layer of LLaMA-65B.

Using the sensitivity analysis, we observe several patterns in the weight matrix, often in a single row or column. Since the large weight matrices in LLaMA-65B have too many rows/columuns to be respresentable in a compact image (default: 8k $\times$ 32k pixels) we perform max pooling to visualize the matrices, that is we take the maximum sensitivity in each square of $32 \times 32$ rows and columns. This max pooling only affects the leftmost image. Using this visualization, we observe that the quantization error patterns vary both by layer type, for example attention vs multilayer perceptron (MLP), and layer depth. In particular, we find that more sensitive outliers are present for deeper layers. (Please see Appendix B for additional results.) To categorize outlier structures, we will use the attention weight matrix as a reference. We make the following observations:

- **Row outliers** are shown in Figure 2 bottom-center as regions of high sensitivity within one output unit. Some of these patterns span the entire row, while others are partial. In attention layers, some of the partial row outliers correspond to some subset of attention heads. **Column outliers** appear in Figure 2, bottom-right, showing high sensitivity in select input dimensions (columns) across all rows. The latter is related to the "outlier feature" phenomenon reported in Dettmers et al. (2022).

- **Sensitive attention heads.** (Figure 2, top-center) – regular stripes of width 128 highlight all weights corresponding to one attention head. This could be related to some attention heads having more important functions (Voita et al., 2019; Vig, 2019; Olsson et al., 2022). The corresponding "stripes" are horizontal for attention Q & K projections, vertical in output projection, and absent from value projections and any MLP weights. Of note, there is significant variation in individual weight sensitivity even within the sensitive heads.

- **The Rotary embedding pattern**, a repeating vertical pattern of sensitivity with a period of 64 units. We attribute this to the use of rotary embeddings (Su et al., 2021): each attention head (dim = 128) is split into two halves: the first 64 are "rotated" with cosine, and the other 64 use sine. Both sine and cosine rotation use the same set of frequencies. Typically, the weights that correspond to low-frequency sines and cosines are more sensitive than their high-frequency counterparts, as shown in Figure 2 (top-right). This pattern is absent from any layer not using rotary embeddings.

- **Unstructured outliers.** Besides the above, each layer has a number of individual sensitivity weights that do not fit into any of the above patterns. These unstructured outliers occur more frequently for columns with largest input index (i.e., on the right side of the images). This effect is difficult to see on a heatmap, so we provide additional figures and statistical tests in Appendix B. We believe is probably an artifact of the GPTQ algorithm, which compresses one by one, using yet-uncompressed weights to compensate the error. Thus, the rightmost batch of weights accumulates the most error.

Next, we will leverage these findings to propose a compressed representation which can support all these different outlier types.

## 4 SpQR: A Sensitivity-aware compressed representation

### 4.1 Overview

Existing LLM quantization algorithms treat low- and high-sensitivity weights equally; however, our above discussion suggests that this may lead to sub-optimal compression. Ideally, we would want the representation to assign more of its "size budget" to sensitive weights. However, these weights are scattered in the weight matrix as either individual weights or small groups, for example, partial rows or attention head. To capture this structure, we are introducing two changes to the quantization procedure: one for capturing small sensitive groups, and another for capturing individual outliers.

**Capturing small groups of weights with bilevel quantization.** In the previous section, we observed several cases where weights behave similarly in small consecutive groups, with abrupt changes between groups, for example for some attention head and partial row outliers (see Figure 2). When applying a standard approach, there will be many cases where these weights will be grouped together, sharing the same quantization statistics. To reduce the number of such cases, we use groupwise quantization with extremely small groups, typically of 8–32 weights. That is, for every $\beta_1$ consecutive weights, there is a separate quantization scale and zero-point. This choice runs contrary to current intuition: for instance, Yao et al. (2023) recommends against small groups, arguing that the overhead for storing quantization statistics would outweigh the precision advantages.

To circumvent this issue, we quantize the groupwise statistics themselves using the same quantization algorithm as for weights — asymmetric (min-max) quantization. In other words, we group groupwise statistics from $\beta_2 = 16$ consecutive values and quantize them together in the same number of bits, such that groups with atypical quantization parameters end up using more of the "quantization budget". Finally, both first and second-level quantization happens directly within the quantization process, allowing the algorithm to compensate the second-level quantization error where possible.

**High-sensitivity outliers.** Our analysis shows the existence of cases where a small percentage of sensitive weights come in small groups (in the self-attention) or individual "outliers" (in the MLP). In some cases, 1% of the weights account for over 75% of the total quantization error. Since these weights appear to lead to high, irreducible error, we choose to keep these outliers in 16-bit precision.

The procedure for detecting the outliers is described in detail in Alg. 1. It relies on **quantize**, **dequantize** and **fit_quantizer** functions that perform standard min-max quantization (see full definition in Appendix H). The algorithm follows a rough two-step procedure: (1) find and isolate outliers as 16-bit weights, (2) quantize the non-outlier "base" weights into 3-4 bit and transfer the remaining

**Algorithm 1** SpQR quantization algorithm: the left snippet describes the full procedure, the right side contains subroutines for bilevel quantization and finding outliers.

```
func SPQRQUANTIZE(W, X, b, β₁, β₂, τ, λ)
```
**Input:** $W \in \mathcal{R}^{m \times n}$ — weight matrix,
$\quad\quad X \in \mathcal{R}^{n \times d}$ — calibration data,
$\quad\quad b$ — the base number of quantization bits,
$\quad\quad \beta_1, \beta_2$ — quantization group sizes,
$\quad\quad \tau$ — sensitivity outlier threshold
$\quad\quad \lambda$ — hessian regularizer,
1: $E := \text{float\_matrix}(m, n)$   // L2 error
2: $H := 2XX^T$   // L2 error hessian, $\mathcal{R}^{n \times n}$
3: $H^{\text{ic}} := \text{Cholesky}((H + \lambda\mathbf{I})^{-1})$
4: $Q := \text{int\_matrix}(m, n)$   // quantized weight
5: $\mathcal{O} := \emptyset$   // a set of all outliers
6: $\mathcal{S} := \emptyset$   // a set of quantization statistics
7: **for** $i = 1, \beta_1, 2\beta_1, \dots n$ **do**
8: $\quad W_{:,i:i+\beta_1}, \mathcal{O} := \text{outliers}(W_{:,i:i+\beta_1}, H^{\text{ic}}_{i:(i+\beta_1),i:(i+\beta_1)}\mathcal{O})$
9: $\quad \hat{s}, \hat{z}, \mathcal{S} := \text{fit\_statistics}(W_{:,i:i+\beta_1}, \mathcal{S}, \mathcal{O})$
10: $\quad$ **for** $j = i, \dots, i + \beta_1$ **do**
11: $\quad\quad Q_{:,j} := \text{quantize}(W_{:,j}, \hat{s}, \hat{z})$
12: $\quad\quad \vec{w}_q := \text{dequantize}(Q_{:,j}, \hat{s}, \hat{z})$
13: $\quad\quad E_{:,j} := (W_{:,j} - \vec{w}_q)/H^{\text{ic}}_{j,j} \cdot (1 - \text{is\_outlier}(W_{:,j}, \mathcal{O}))$
14: $\quad\quad W_{:,j:(i+\beta_1)} := W_{:,j:(i+\beta_1)} - E \cdot H^{\text{ic}}_{j,j:(i+\beta_1)}$
15: $\quad W_{:,(i+\beta_1):n} := W_{:,(i+\beta_1):n} - E \cdot H^{\text{ic}}_{i:(i+\beta_1),i:(i+\beta_1)}$
16: $S_q, Z_q, S_s, Z_s, S_z, Z_z := \text{gather\_statistics}(\mathcal{S})$
17: $W_{sparse} = \text{gather\_outlier\_matrix}(W, \mathcal{O})$
18: **return** $Q, S_q, Z_q, S_s, Z_s, S_z, Z_z, W_{sparse}$

```
func error(W, H^ic)
```
1: $\vec{s}, \vec{z} := \text{fit\_quantizer}(W, \beta_1)$
2: $W_q := \text{quantize}(W, \vec{s}, \vec{z})$
3: $E := (W - W_q)/H^{\text{ic}}$
4: **return** $E^2$

```
func outliers(W, H^ic, O)
```
1: $E_{\text{base}} = \text{error}(W, H^{\text{ic}})$
2: **for** $i = 1, \dots, \beta_1$ **do**
3: $\quad loo := \{1, 2, \dots, \beta_1\}/\{i\}$
4: $\quad E_{\text{ol}} = \text{error}(W_{:,\text{loo}}, H^{\text{ic}}_{\text{loo,loo}})$
5: $\quad I_o = \text{select}(E_{\text{base}} - E_{\text{ol}} > \tau)$
6: $\quad \mathcal{O} := \mathcal{O} \cup I_o$
7: **return** $W, \mathcal{O}$

```
func fit_statistics(W, S, O)
```
1: $W := W \cdot (1 - \text{is\_outlier}(W, O))$
2: $\vec{s}, \vec{z} := \text{fit\_quantizer}(W, \beta_1)$
3: // $\vec{s}$ for scales, $\vec{z}$ for zero points
4: $\vec{s}_s, \vec{z}_s := \text{fit\_quantizer}(\vec{s}, \beta_2)$
5: $\vec{s}_z, \vec{z}_z := \text{fit\_quantizer}(\vec{z}, \beta_2)$
6: $\vec{s}_q := \text{quantize}(\vec{s}, \vec{s}_s, \vec{z}_s)$
7: $\vec{z}_q := \text{quantize}(\vec{z}, \vec{s}_z, \vec{z}_z)$
8: $\mathcal{S} := \mathcal{S} \cup \{s_q, s_s, s_z, z_q, s_z, z_z\}$
9: $\hat{s} := \text{dequantize}(s_q, s_s, s_z)$
10: $\hat{z} := \text{dequantize}(z_q, s_z, z_z)$
11: **return** $\hat{s}, \hat{z}, \mathcal{S}$

quantization into the the 16-bit outliers weights. For the outlier isolation step, the algorithm implements a filtering technique based on the sensitivity criterion in defined in Section 3.1, which is used to isolate and separate outliers from base weights. We pick a model-dependent sensitivity threshold $\tau$ to obtain the desired number of outliers across the whole model. Usually, the percentage of outliers as a share of total parameters has good memory/performance trade-off at around 0.3% to 1% of outlier weights which translates to $\tau \in [0.1, 0.40]$.

Following this first outlier detection step, we quantize the base weights ignoring all outliers that occur in the same quantization group. As such, the quantization statistics (e.g., scales) are computed by excluding outliers. This results in significant improvements in terms of error, since e.g. the min-max scales will be significantly reduced. Interestingly, unlike Dettmers et al. (2022), a weight can be chosen to be an outlier not only if it causes error by itself, but also if the GPTQ algorithm can employ this weight to compensate errors from many other weights. Thus, the resulting 16-bit value will contain not the original weight, but a weight that was adjusted to minimize the output error. As such, SpQR goes beyond mere detection of outliers towards the more general notion of isolating and treating outliers that occur *during* the quantization process. Finally, the algorithm gathers and compresses sparse outlier matrix as well as the final quantization statistics with bilevel quantization and returns the compressed weights and their metadata.

## 4.2 STORING AND INFERENCING ON SPQR MODELS

Our algorithm converts homogeneous weights into different data structures of various sizes and precisions, as depicted in Figure 3. Overall, the representation consists of four parts: (1) individual weight codes, (2) first level quantized quantization statistics, (3) second level quantization statistics, and (4) the sparse outlier indices and values. The first three parts are regular dense tensors that are easy to store and inference. In turn, the sparse outliers present a bigger challenge.

A naive way to encode a sparse outlier matrix would be to use compressed sparse row (CSR) format. This format groups outliers by their row index. In each row, CSR stores pairs of outlier value (e.g. float16) and it's column index (e.g. int16). In this format, at least half of the bits are spent on representing outlier locations, not their values. To reduce this overhead, we replace outlier indices with **relative index shifts between the adjacent outliers**. Instead storing a 16-bit index $I_k$, we could store 8-bit difference $\Delta_k = I_k - I_{k-1}$, but only if adjacent outliers are at most 255 columns apart.

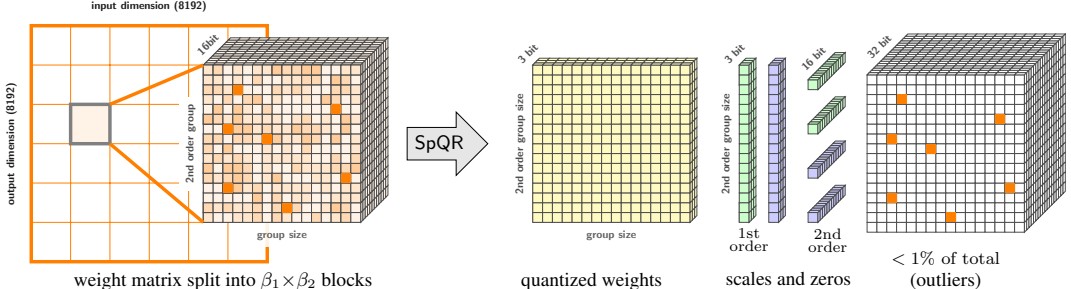

Figure 3: A high-level overview of the SpQR representation for a single weight tensor. The right side of the image depicts all stored data types and their dimensions.

In practice, not all outliers are within this range: about $3-7\%$ of those outliers would not fit into 8-bit $\Delta_k$. Whenever adjacent outliers are too far away, *we introduce an additional "virtual" outlier with zero value*. Virtual outliers have no effect on model behavior, but they allow storing all $\Delta_k$ shifts as 8-bit integers. This results in an average of $8-9$ bits per *real* outlier index, compared to 16 bits in CSR. When running GPU inference in this format, we can recover the outlier indices on the fly using the parallel prefix sum (scan) algorithm (Harris et al., 2007). The resulting format allows for a more memory-efficient outlier representation with negligible effect on GPU inference speed. This format starts being more effective than absolute indices at around $0.1\%$ of the total number of weights.

**Inference with SpQR.** To illustrate the practicality of our approach, we design an efficient GPU-based decoding implementation for the SpQR format, focused on the popular token-by-token LLM generation as a use-case. We leverage the fact that autoregressive inference on GPUs is memory-bound, so high compression rates can hide decoding overheads, to a significant extent. At a high level, our algorithm loads group statistics and the quantized weights into shared memory (SRAM), dequantizes to 16-bits, and then performs matrix multiplication with 16-bit inputs. For handling outliers, we design a sparse matrix algorithm that takes advantage of outliers that occur in rows. Roughly, the algorithm works as follows. First, (1) we divide the matrix into equally sized blocks. Then, each GPU core (thread block) (2) loads a large slice of outliers into shared memory (SRAM), and each GPU core (3) determines if outliers are part of the segment or not. The corresponding weights are (4) loaded from main memory; finally, the matrix multiplication is performed.

This algorithm essentially performs load balancing through steps (1-3), while step (4) tends to have contiguous memory access due to the row-like patterns for the outliers. We will show in Section 5 that this custom approach is faster than the sparse matrix algorithms in PyTorch.

## 5 EXPERIMENTAL VALIDATION

**Experimental setup.** We focus on three main goals: 1) evaluating most compact representation with which SpQR can replicate the performance of a 16-bit model within 1% perplexity, 2) controlling for the average number of bits per parameter across methods and compare to round-to-nearest (RTN) and GPTQ baselines, 3) finding the best trade-off in terms of model size and performance. For these settings, we evaluate the full SpQR algorithm on publicly-available LLMs. We focus on the LLaMA-$\{7, 13, 30, 65\}$B model family (Touvron et al., 2023) and Falcon-$\{7, 40, 180\}$B model family (TII UAE, 2023a). We quantize LLaMA models using the RedPajama dataset and Falcon models on RefinedWeb datasets (TII UAE, 2023b), both of which represent the original training data or open replications of the training data. We provide additional results, including for Llama 2 and OPT models, in Appendix I.

We compare SpQR against two other post-training quantization schemes: GPTQ (Frantar et al., 2022a) and simple rounding-to-nearest (RTN) quantization, which is used by most other LLM compression methods (Dettmers et al., 2022; Yao et al., 2022). Both baselines use 4-bit quantization since it provides the best quality to size trade-off (Dettmers & Zettlemoyer, 2022). For SpQR, we consider both 3-bit and 4-bit base quantization, though the resulting model size is slightly larger due to outliers.

We evaluate quantized model performance by two metrics. Firstly, we measure *perplexity*, measured on the WikiText2 (Merity et al., 2016), Penn Treebank (Marcus et al., 1994) and C4 (Raffel et al., 2020) datasets. Secondly, we measure zero-shot accuracy on five tasks: WinoGrande (Sakaguchi et al., 2021), PiQA (Tata & Patel, 2003), HellaSwag, ARC-easy and ARC-challenge (Clark et al., 2018). We use the LM Evaluation Harness (Gao et al., 2021) with recommended parameters. We provide full configurations in Appendix C, as well as code which we plan to release publicly. Our

Table 1: Perplexity on WikiText2 (Merity et al., 2016), C4 (Raffel et al., 2020) and Penn Tree-bank (Marcus et al., 1994) for SpQR and round-to-nearest (RTN) and GPTQ baselines on LLaMA and Falcon models. We can see that SpQR reaches performances within 1% of the perplexity with less than 4.5 bits per parameter. We also see that for 4-bits per parameter SpQR significantly improves on GPTQ with an improvement as large as the improvement from RTN to GPTQ.

**Falcon**

| Size | Method | Avg bits | Wiki2 | C4 | PTB |
|---|---|---|---|---|---|
| | – | 16.00 | 6.59 | 9.50 | 9.90 |
| | SpQR | 4.44 | 6.64 | 9.58 | 9.97 |
| 7B | RTN | 4 | 8.73 | 12.56 | 13.76 |
| | GPTQ | 4 | 6.91 | 9.93 | 10.33 |
| | SpQR | 3.92 | 6.74 | 9.70 | 19.114 |
| | – | 16.00 | 5.23 | 7.76 | 7.83 |
| | SpQR | 4.46 | 5.26 | 7.79 | 7.86 |
| 40B | RTN | 4 | 6.52 | 9.76 | 10.63 |
| | GPTQ | 4 | 5.36 | 7.95 | 8.01 |
| | SpQR | 3.90 | 5.29 | 7.85 | 7.91 |
| | – | 16.00 | 3.30 | 6.37 | 6.65 |
| 180B | GPTQ | 4 | 3.66 | 6.83 | 6.58 |
| | SpQR | 3.83 | 3.43 | 6.41 | 6.71 |

**LLaMA**

| Size | Method | Avg bits | Wiki2 | C4 | PTB |
|---|---|---|---|---|---|
| | – | 16.00 | 5.68 | 7.08 | 8.80 |
| | SpQR | 4.63 | 5.73 | 7.13 | 8.88 |
| 7B | RTN | 4 | 6.43 | 7.93 | 10.30 |
| | GPTQ | 4 | 6.13 | 7.43 | 9.27 |
| | SpQR | 3.94 | 5.87 | 7.28 | 9.07 |
| | – | 16.00 | 4.10 | 5.98 | 7.30 |
| | SpQR | 4.69 | 4.14 | 6.01 | 7.33 |
| 30B | RTN | 4 | 4.57 | 6.34 | 7.75 |
| | GPTQ | 4 | 4.48 | 6.20 | 7.54 |
| | SpQR | 3.89 | 4.25 | 6.08 | 7.38 |
| | – | 16.00 | 3.53 | 5.62 | 6.91 |
| | SpQR | 4.71 | 3.57 | 5.64 | 6.93 |
| 65B | RTN | 4 | 3.87 | 5.85 | 7.17 |
| | GPTQ | 4 | 3.83 | 5.80 | 7.07 |
| | SpQR | 3.90 | 3.68 | 5.70 | 6.99 |

implementation takes around 4.5 hours on the largest model size (65B) on an NVIDIA A100 (80 GB). Our emory efficient implementations take 12 hours on a small 24 GB GPU.

To control for model size, we evaluate RTN and GPTQ with 4-bit base quantization. For SpQR we use 3-bit base quantization, a group size of 8 with 3-bit for the first quantization, a group size of 64 for the second quantization, and as many outliers as possible to still reach less than 4-bits per parameter on average. We aim to achieve *near-lossless* compression, for which we adopt the definition of the MLCommons benchmark (Reddi et al., 2020): *1% error relative to the uncompressed baseline*. In all SpQR evaluations, we choose $\tau$ such that the proportion of outliers is under $1\%$.

**Main Results.** Figure 1 measures model size versus perplexity on LLaMA models on WikiText2, and accuracy on zero-shot tasks. We observe that SpQR outperforms GPTQ (and correspondingly RTN) at similar model size by a significant margin, especially on smaller models. This improvement comes from both SpQR achieving more compression, while also reducing loss degradation. In addition, if we measure the bits per parameter needed to come within 1% of the 16-bit performance in terms of perplexity, Figure 1 shows that SpQR with 4.6 to 4.71 bits per parameter approaches the non-quantized models with at most 1% margin of error for all models (see Table 1 for exact values).

Additional results where we control overall model size and compare quantization methods at an average of 4-bits per parameter are presented in Table 1. We see that SpQR improves over previous methods, with the gap between SpQR and the next best method GPTQ being as large as the improvement of GPTQ over naive RTN. For 4-bit, SpQR *halves the error* relative to the 16-bit baseline compared to GPTQ. We report evaluations for Llama 2 and OPT models in the Appendix I.

**Ablations.** The SpQR representation differs from standard quantization methods in two main ways: bilevel quantization with small quantization group size and unstructured outliers. To understand the effect of small group sizes, we compare 3-bit SpQR with group size 16, compressed using 3-bit bilevel quantization, versus a setup with group size 48, keeping quantization statistics in 16-bit. Both configurations result in approximately 3.6 average bits per parameter. For simplicity, neither uses outliers. We report both in Table 2, the "3-bit statistics" entry corresponds to group size 16 with 3-bit statistics and "16-bit statistics" stands for group size 16 with 16-bit statistics. Given the same (slightly smaller) memory footprint, using quantized statistics significantly improves language modeling loss.

Next, we ask whether it is necessary to use unstructured outliers, considering two outlier types. First, we use the criterion of Dettmers & Zettlemoyer (2022) to find column outliers and quantize them in

| Name | Wiki2 | C4 | PTB | Avg bits |
|---|---|---|---|---|
| Uncompressed | 3.53 | 5.62 | 6.91 | 16 |
| GPTQ (4 bit) | 3.83 | 5.80 | 7.07 | 4 |
| 3-bit statistics | 3.74 | 5.73 | 7.02 | 3.63 |
| 16-bit statistics | 3.84 | 5.83 | 7.12 | 3.67 |
| Round zero | 3.75 | 5.76 | 7.01 | 3.63 |
| w/o act order | 3.74 | 5.76 | 7.05 | 3.63 |

Table 2: Perplexity for LLaMA-65B model.

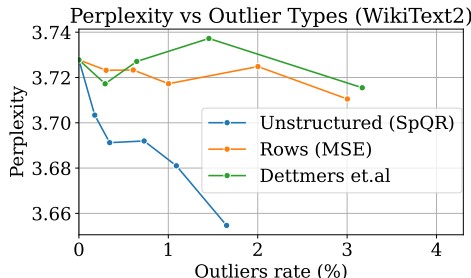

Figure 4: Different outlier types, LLaMA-65B.

higher precision. The alternative is to treat the entire rows (output units / hidden units / neurons) as outliers: we run SpQR without outliers, then select $k$ output units that have the highest quantization error (i.e., MSE between layer predictions) and treat the entire rows as 16-bit outliers. We compare the three outlier types on top of 3-bit SpQR and report the results in Figure 4. Overall, unstructured outliers reduce perplexity significantly faster than their row counterpart and the criterion of Dettmers & Zettlemoyer (2022), even after accounting for the different memory footprint.

Finally, we analyze the impact of the minor hyperparameter changes that we introduced at the end of Section 4. In Table 2 (bottom), we evaluate quantization errors without these changes. The "Round zero" entry corresponds to a version of SpQR where the zero-point is a 3-bit integer. This reduces the memory footprint of SpQR, but results in a moderate increase in perplexity. Similarly, we evaluate SpQR without the "act order" flag. This option re-orders the input dimensions by the diagonal of the inverse hessian, which was introduced as a part of the GPTQ algorithm. Using this heuristic slightly improves loss, though not as much as from quantized groups. To further explore the design space of SpQR, we provide an additional hyperparameter study in Appendix D.

**Inference Time.** Finally, we evaluate LLM inference speed of SpQR for batch size 1 on a single A100 GPU. We measure inference speed in two setups: i) generating 100 tokens from scratch and ii) adding 100 tokens on top of a 1024-token prefix (prompt). We compare our sparse matrix multiplication algorithm with the PyTorch default (cuSPARSE). We also compare against a 16-bit baseline. We measure the end-to-end latency as inference steps per second for the full SpQR algorithm, that is for both the dense and sparse multiplication part together. Results are shown in Table 3, where we see that our specialized sparse matrix multiplication algorithm yields speedups of about 20-30% and is 2x faster than a PyTorch implementation.

Table 3: Inference speed comparison (tokens/s), OOM means the model did not fit in an A100 GPU. We see that our optimized SpQR algorithm is faster than the 16-bit baseline and almost 2.0x faster than quantized matrix multiplication + standard PyTorch sparse matrix multiplication baseline.

| Method | fp16 (baseline) | | | | SpQR (PyTorch) | | | | SpQR (optimized) | | | |
|---|---|---|---|---|---|---|---|---|---|---|---|---|
| LLaMA | 7B | 13B | 30B | 65B | 7B | 13B | 30B | 65B | 7B | 13B | 30B | 65B |
| scratch | $47 \pm 2.3$ | $37 \pm 0.8$ | $19 \pm 1.1$ | OOM | $30 \pm 2.2$ | $24 \pm 1.2$ | $8.8 \pm 0.4$ | OOM | $\mathbf{57} \pm 2.4$ | $\mathbf{44} \pm 0.5$ | $\mathbf{22} \pm 0.9$ | $\mathbf{12} \pm 0.6$ |
| prefix 1024 | $46 \pm 2.4$ | $31 \pm 0.9$ | $17 \pm 0.8$ | OOM | $27 \pm 1.6$ | $21 \pm 1.1$ | $6.5 \pm 0.7$ | OOM | $\mathbf{55} \pm 2.1$ | $37 \pm 0.8$ | $\mathbf{22} \pm 1.3$ | $\mathbf{11} \pm 0.6$ |

# 6 DISCUSSION & LIMITATIONS

We have presented SpQR, a compression approach which quantizes sensitive outliers in higher precision, to achieve near-lossless 16-bit accuracy with less than 4.75 bits per parameter on average. We achieve even better quality-size-tradeoffs when compressing to as little as 3.5 bits. This makes SpQR an ideal method for compressing models for memory-limited devices. Despite our promising results, there are a few limitations. The main limitation is that we do not evaluate the generative quality of quantized LLMs, but only the predictive performance in terms of zero-shot accuracy and perplexity. While we believe that perplexity measurements and generation quality are strongly related, this is a hypothesis we aim to investigate in future work. While we devise a sparse matrix multiplication algorithm to accelerate the computation with outliers, another limitation is that we do not fuse sparse matrix multiplication with regular quantized matrix multiplication. Such an approach would yield even better inference time performance. We plan to investigate such practical kernel extensions more generally in future work.

## 7 ACKNOWLEDGEDMENTS

Denis Kuznedelev acknowledges the support from the Russian Ministry of Science and Higher Education, grant No. 075-10-2021-068. Ruslan Svirschevski and Vage Egiazarian and Denis Kuznedelev were supported by the grant for research centers in the field of AI provided by the Analytical Center for the Government of the Russian Federation (ACRF) in accordance with the agreement on the provision of subsidies (identifier of the agreement 000000D730321P5Q0002) and the agreement with HSE University No. 70-2021-00139.

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

TABLE OF CONTENTS

## A  ADDITIONAL RELATED WORK

**Hybrid sparse-quantized formats**    Hybrid sparse-quantized formats have been investigated generally for deep networks. Some efficient CPU inference engines (NeuralMagic, 2022; Gorbachev et al., 2019) support a different block sparse-and-quantized format, in which each block of 4 consecutive weights is either completely sparse or quantized to 8-bit format, whereas GPUs support a similar compound format in which every group of 4 weights contains 2 zero weights, and the non-zero weights could be quantized. The FBGEMM package (Khudia et al., 2021) proposed a format in which certain "outlier" weights are quantized separately, to reduce their impact on normalization. However, in this format, "outlier" weights are still quantized to exactly the same bit-width (8-bit) as regular weights; moreover, no procedure is given for converting a model to this format post-training. By contrast, 1) we provide an efficient and accurate post-training compression algorithm which identifies outliers as weights inducing high output error, 2) we propose a format compressing outliers to a higher bit-width relative to regular weights, and 3) our format stores outliers in blocks, allowing for efficient implementation of GPU kernels, which we provide as well.

**Concurrent work on LLM compression** Several recent works have studied the compression of large language models via quantization. One of the challenges of LLM quantization is the presence of outlier weights (Dettmers et al., 2022), that are very sensitive to quantization or may significantly increase quantization range thus leading to inaccurate representation of other weights. To address this problems several approachers were proposed. AWQ (Lin et al., 2023) skips quantization of channels with largest magnitude of activations and performs per-channel scaling to protect salient weights. SqueezeLLM (Kim et al., 2023) adopts diagonal Fisher as a Hessian proxy and applies non-uniform quantization with K-means clustering. This works define two types of outliers - those sensitive to quantization with respect to the Hessian approximation and with large absolute weight value. These outliers are kept in original floating point precision. We emphasize, that these approaches while solving the same problem are quite different from the introduced SpQR method. See section G for comparison.

Other approaches seek to learn distributions during training which are more amenable to quantization (Strom et al., 2022) or by suppressing outliers (Ahmadian et al., 2023; Wortsman et al., 2023).

# B  ADDITIONAL WEIGHT SENSITIVITY ANALYSIS

In this section, we provide additional visualizations of LLaMA weight sensitivities, as well as additional plots for different layer roles. As we observed earlier in Section 3.2, the sensitivity matrices vary based on four main factors:

- the quantization scheme (e.g. row- or group-wise);
- the layer depth, i.e. the index of the corresponding transformer block;
- the role of that weight, e.g. self-attn query / key or MLP up / down projection;
- the location within the chosen weight matrix;

Here, we report additional observations about these factors and elaborate on some of our claims from Section 3.1. We also report raw sensitivity matrices for various weight matrices at the end of the supplementary materials.

**Relation between sensitivity and the chosen quantization scheme.** We compare two configurations of GPTQ 3-bit. The first configuration uses one quantization scale & zero for each row. The second one uses blockwise quantization with one set of statistics for each block of 128 weights.

Figure 5 demonstrates a typical example of how group size affects sensitivity. In the bottom-right plot, we observe that a subset of weights (width 128) has a significantly higher quantization error than the rest of the layer. Please note that the color scale represents sensitivity on a logarithmic scale, with higher sensitivity being darker.

On a more detailed examination, we found that this specific group contains a "vertical" outlier, i.e. the corresponding input feature has significantly higher variance, compared to other input dimensions.

In this example, the main effect of GPTQ block size 128 is that the problematic dimension leads to increased sensitivity in a group of $8192 \times 128$ weights. In turn, GPTQ with per-row statistics has high quantization error across the entire row.

**The effect of rotary embeddings.** Earlier in Figure 2 we note that attention query and key have a regular pattern of sensitivity that repeats every 64 rows. We attribute this to the fact that LLaMA uses rotary position embeddings. More specifically, this pattern is likely a side-effect of how rotary embeddings are implemented for this model.

To recall, rotary position embeddings are a technique that rotates attention head dimensions by an angle that depends on how many tokens are between key and query (Su et al., 2021). Furthermore, dimensions within each head are rotated with a different frequency. To implement this rotation, LLaMA multiplies each head by a precomputed tensor of sine and cosine functions with a different period. The first half (64 units) of the matrix is multiplied by cosines and the other half (64 units) is multiplied by sines.

To recall, sine and cosine components are equivalent up to a phase shift and show similar behavior in our analysis. In general, we observe that weights that correspond to low-frequency heads (bottom of

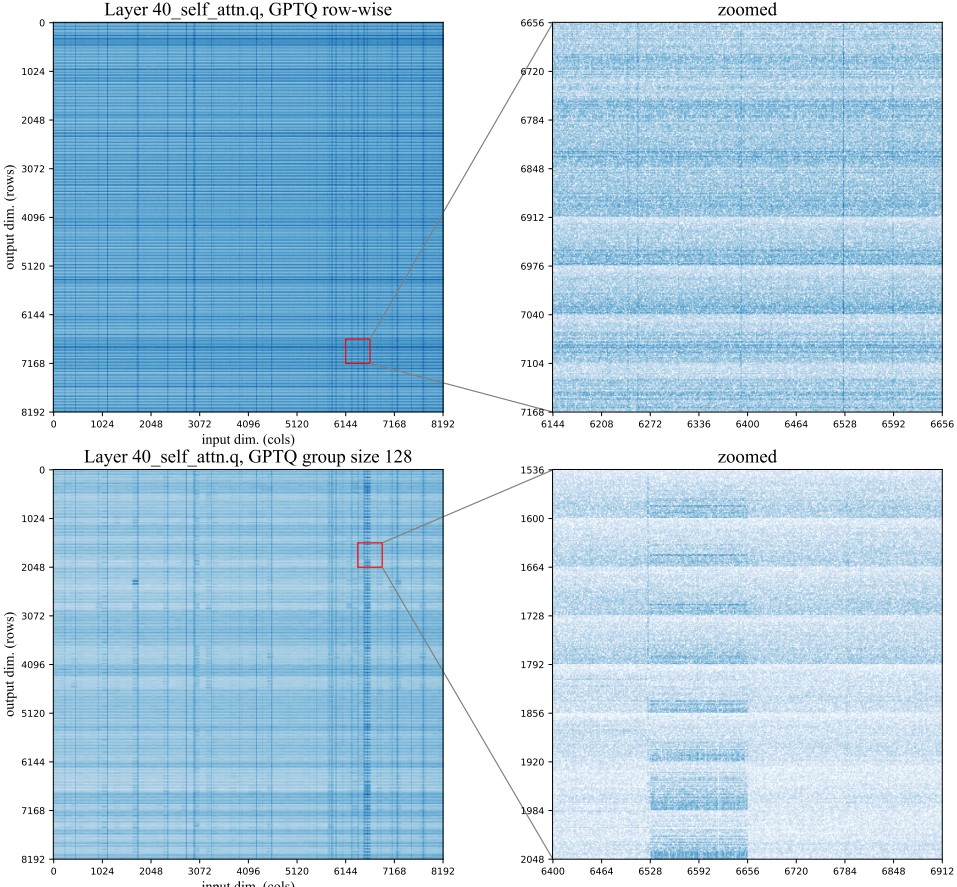

Figure 5: The weight sensitivities for LLaMA-65B 40th layer, attention query projection. The color scale represents sensitivity on a logarithmic scale, with higher sensitivity being darker. **(top)** 3-bit GPTQ with per-row quantization scales, **(bottom)** 3-bit GPTQ with block size 128.

each semi-head) typically have higher sensitivity. One possible explanation is that high-frequency heads can be more dependent on position-specific information, such as attending to the previous token — and less dependent on the weights that represent content information. However, this phenomenon merits further investigation and our current understanding should be treated as an educated guess.

**GPTQ and the effect of quantization order.** As we observe earlier in Section 3.2, the rightmost weights in each visualization tend to have higher quantization errors. This is likely a side-effect of the GPTQ algorithm, which compresses weights one input feature at a time, i.e. column by column in a left-to-right direction. Once a column is quantized, the algorithm uses the remaining unquantized weights to compensate for the error. Thus, the rightmost batch of weights accumulates the most error from preceding columns and has the least space to compensate it's "own" quantization error.

This difference is most pronounced in the earlier layers, where the quantization error is smaller overall (see Figure 6). To further verify this observation, we observe that this effect disappears if we shuffle the weight quantization order in the GPTQ algorithm.

**Relation between weight sensitivity and layer depth.** In terms of mean squared error, we observe that the first layers of LLaMA tend to have generally lower OBC error (defined as L2 distance between original and quantized layer predictions). To illustrate this, we report the average quantization error of GPTQ-3bit in Figure 7.

The absolute quantization error means little by itself since each quantized layer has a different input/output variance. However, we also observe that the first and last few layers have qualitative differences in behavior. Figures 11 and 12 report weight sensitivities for the first, middle (40th), and last (79th) layer of LLaMA model separately to better illustrate this difference.

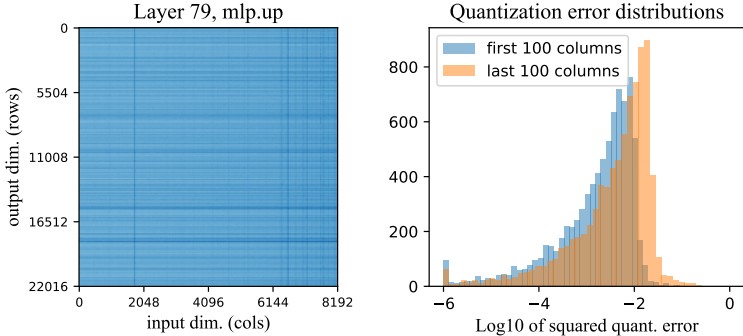

Figure 6: The weight log-sensitivities for a deeper upward projection layer (in particular, this is layer #79). The heatmap on the left represents the sensitivities of each weight, with darker being more sensitive; the histogram on the right captures the sensitivities in the first 100 and last 100 columns (sorted across input dimensions). The latter figure clearly shows that later columns are more sensitive on average.

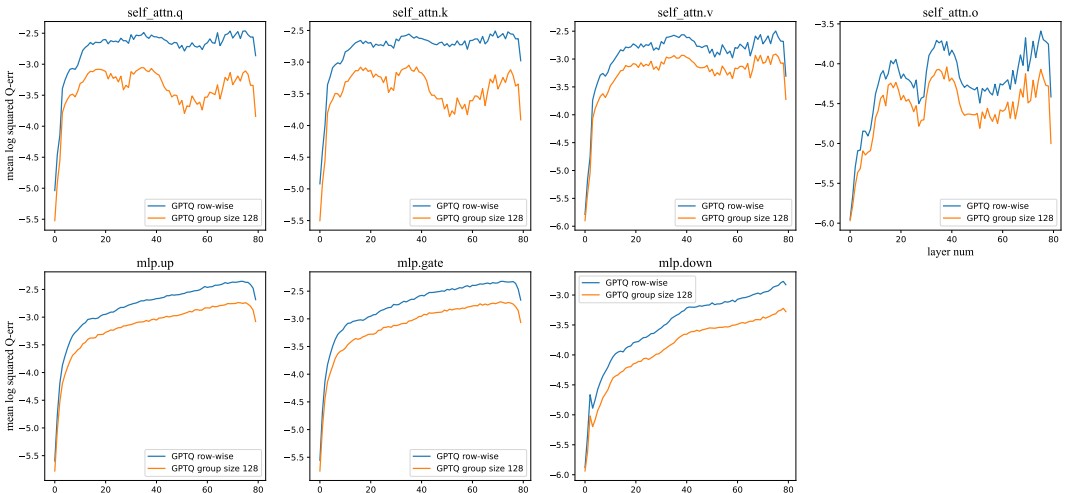

Figure 7: Mean quantization error (vertical axis) as a function of layer depth (horizontal axis). Each plot corresponds to a different layer role.

## C  EXPERIMENTAL CONFIGURATIONS

The SpQR representations proposed in this work have several adjustable hyperparameters that allow for great flexibility in targeting a desired size of the model. We introduce the notation and list the method hyperparameters below:

- $b_w$ - number of bits per weight
- $b_s$ - number of bits per scale
- $b_z$ - number of bits per zero
- $r_o$ - outlier rate (fraction of weights that are not quantized)
- $\beta_1$ - block size for weight quantization
- $\beta_2$ - block size for statistic quantization;
- $\tau$ - outlier threshold

The actual number of outliers depends not only on $\tau$, but on all other hyperparameters as well. However, for any specific configuration, increasing $\tau$ leads to reduced number of outliers. To achieve the desired number of outliers, we tune $\tau$ in $[0.1, 1.0]$ range by binary search with minumum step size 0.05. The vast majority of our configurations are between $\tau = 0.1$ and $\tau = 0.45$].

The full configuration we use to compress LLaMA-30B model near-losslessly in Table 1 has the following hyperparameters: $b_w = 4, b_s = b_z = 3, \beta_1 = \beta_2 = 16, \tau = 0.1$ This translates to the following command line arguments in our supplementary code:

```
python main.py $MODEL custom --custom_data_path=$DATA  \
    --wbits 4 --groupsize 16 --perchannel --qq_scale_bits 3  \
    --qq_zero_bits 3 --qq_groupsize 16 --outlier_threshold 0.1 \
    --fit_quantizer_without_outliers --permutation_order act_order
```

We used Weights & Biases  Biewald (2020) to track our experiments.

## D  HYPERPARAMETER SENSITIVITY

In this section, we analyze how SpQR performance depends on the choice of quantization group sizes. Please recall that the SpQR algorithm uses two types of groups, indexed by parameters $\beta_1$ and $\beta_2$. The first group dimension $\beta_1$ covers multiple weights for the same input unit, similar to standard blockwise quantization. In turn, the other dimension $\beta_2$ covers multiple output units, and is used when quantizing quantization scales. In our visualizations, $\beta_1$ blocks are always horizontal, while $\beta_2$ are vertical.

In Table 4, we evaluate SpQR with varying parameters $\beta_1$ and $\beta_2$. We quantize LLaMA-65B with 3-bit SpQR for weights and statistics and report perplexity on WikiText2, Penn Treebank, and C4 datasets. The upper-left section of the table contains the effective number of bits for each group configuration, and the remaining sections correspond to perplexities on different datasets.

To summarize, both small quantized groups and unstructured outliers independently improve perplexity and perform better than alternative strategies. SpQR also benefits from using the GPTQ activation order heuristic, though the gain is smaller than from outliers or small groups. Still, we opt to use the same activation order heuristic in the GPTQ baselines to ensure a fair comparison.

Table 4: Weight block size $\beta_1$ and statistic block size $\beta_2$ performance on WikiText2, C4, and Penn Treebank (PTB). The uncompressed baseline value is provided in the corresponding heading.

| $\beta_1$ \ $\beta_2$ | Average bits | | | | | | Wikitext2 Perplexity (3.53) | | | | | |
|---|---|---|---|---|---|---|---|---|---|---|---|---|
| | 4 | 8 | 16 | 32 | 64 | 128 | 4 | 8 | 16 | 32 | 64 | 128 |
| 4 | 8.5 | 6.5 | 5.5 | 5 | 4.75 | 4.625 | 3.581 | 3.628 | 3.715 | 3.822 | 4.003 | 4.23 |
| 8 | 5.75 | 4.75 | 4.25 | 4 | 3.875 | 3.813 | 3.625 | 3.64 | 3.649 | 3.666 | 3.688 | 3.713 |
| 16 | 4.375 | 3.875 | 3.625 | 3.5 | 3.438 | 3.406 | 3.701 | 3.71 | 3.728 | 3.726 | 3.739 | 3.741 |
| 32 | 3.688 | 3.438 | 3.313 | 3.25 | 3.219 | 3.203 | 3.803 | 3.797 | 3.812 | 3.812 | 3.815 | 3.85 |
| 64 | 3.344 | 3.219 | 3.156 | 3.125 | 3.109 | 3.102 | 3.884 | 3.901 | 3.907 | 3.899 | 3.928 | 3.926 |
| 128 | 3.172 | 3.109 | 3.078 | 3.063 | 3.055 | 3.051 | 3.982 | 3.994 | 4.005 | 3.992 | 4.017 | 4.013 |

| $\beta_1$ \ $\beta_2$ | C4 Perplexity (5.62) | | | | | | PTB Perplexity (6.91) | | | | | |
|---|---|---|---|---|---|---|---|---|---|---|---|---|
| | 4 | 8 | 16 | 32 | 64 | 128 | 4 | 8 | 16 | 32 | 64 | 128 |
| 4 | 5.652 | 5.674 | 5.718 | 5.796 | 5.919 | 6.119 | 6.934 | 6.965 | 7.001 | 7.054 | 7.194 | 7.395 |
| 8 | 5.683 | 5.688 | 5.696 | 5.703 | 5.709 | 5.718 | 6.962 | 6.98 | 6.991 | 6.99 | 6.979 | 7.029 |
| 16 | 5.735 | 5.735 | 5.735 | 5.738 | 5.741 | 5.749 | 7.018 | 7.013 | 7.015 | 7.016 | 7.012 | 7.03 |
| 32 | 5.793 | 5.789 | 5.792 | 5.796 | 5.794 | 5.802 | 7.042 | 7.053 | 7.083 | 7.043 | 7.069 | 7.083 |
| 64 | 5.857 | 5.859 | 5.858 | 5.866 | 5.863 | 5.866 | 7.084 | 7.129 | 7.137 | 7.118 | 7.137 | 7.12 |
| 128 | 5.932 | 5.931 | 5.935 | 5.939 | 5.944 | 5.936 | 7.185 | 7.197 | 7.232 | 7.234 | 7.217 | 7.199 |

## E   ESTIMATING MODEL SIZE

In this section, we provide a quick way to estimate the compressed model size before running the quantization. We express this estimate in terms of *average bits per parameter* defined as:

$$\bar{b} = \frac{\text{model size in bits}}{\text{number of parameters}} \qquad (1)$$

Where model size in bits denotes the total amount of memory - the quantized weights, 1st-order and 2nd-order quantization statistics, outliers and the outlier index - required for the storage of the model. According to Section 4.2, each outlier requires memory storage of $\sim 32$ bits.

The storage and computational cost in transformer models are dominated by the linear projections in the attention and feedforward blocks. Consider quantization of a weight matrix (any of these) $\mathbb{R}^{d_{\text{out}} \times d_{\text{in}}}$ with input dimension $d_{\text{in}}$ and output dimension $d_{\text{out}}$. Then the average number of bits for a given configuration is:

$$\bar{b} \simeq \frac{b_w d_{\text{out}} d_{\text{in}} + (b_s + b_z)\frac{d_{\text{out}} d_{\text{in}}}{\beta_1} + 2(16 + 16)\frac{d_{\text{out}} d_{\text{in}}}{\beta_1 \beta_2}}{d_{\text{out}} d_{\text{in}}} + 32 r_o = b_w + \frac{b_s + b_z}{\beta_1} + \frac{64}{\beta_1 \beta_2} + 32 r_o \quad (2)$$

Therefore, to increase (decrease) the size of the model one should either increase (decrease) the precision of model weights and quantization statistics or decrease (increase) the block size.

For example, for configuration with $b_w = 3, b_s = 3, b_z = 3, \beta_1 = 16, \beta_2 = 32$ and $0.4\%$ of outliers, the average number of bits is:

$$3 + \frac{3 + 3}{16} + \frac{64}{16 \cdot 32} + 0.004 \cdot 32 \simeq 3.63$$

## F   CHOICE OF OPTIMAL CONFIGURATION FOR FIXED AVERAGE NUMBER OF BITS

As discussed above our method has multiple options for improvement of model performance at the cost of the increase of the model size: number of bits per weight $w_b$, groupsizes $b_1$ and $b_2$ for 1st and 2nd order quantization and the outlier rate. We evaluated several configurations with various options for the aforementioned parameters on perplexity benchmarks. Results are presented on Figure 8. One can observe that small groups and small fraction of outliers allows to considerably improve model performance, but the gain is diminishing with the number of bits added (when the additional budget from small group is of order 0.1-0.5 of bits per parameter). It is better to store weights in higher precision instead of keeping them in lower precision but with very small groups or keeping large fraction of outliers. In our experiments optimal fraction of outliers is 0.2-0.5% depending on the model and groupsize.

## G   COMPARISON WITH RECENT CONCURRENT WORK

**Comparison with QuIP at low bitwidth.**   Chee et al. (2023) obtained stable results for 2-bit quantization with the perplexity of the same order of magnitude as the one of the original floating point models, whereas RTN and GPTQ baselines ruin the model completely. SpQR can get stable results in low-bitwidth regime as well, outperforming QuIP results for OPT models with >1B parameters. Nevertheless, we note, that 2-bit quantization is not Pareto-optimal due to significant drop in model performance and for same memory budget usually 4-bit quantization of 2x smaller model is a better option than 2-bit quantization.

**Comparison with AWQ and SqueezeLLM.**   Below we present comparison of SpQR with recent PTQ methods AWQ (Lin et al., 2023) and SqueezeLLM (Kim et al., 2023) on quantization of LLaMA model family at the budget of 3.25 and $\simeq 4$ bits per parameter. The baseline numbers are adopted from Kim et al. (2023). While all methods achieve very similar perplexities, one can see that for

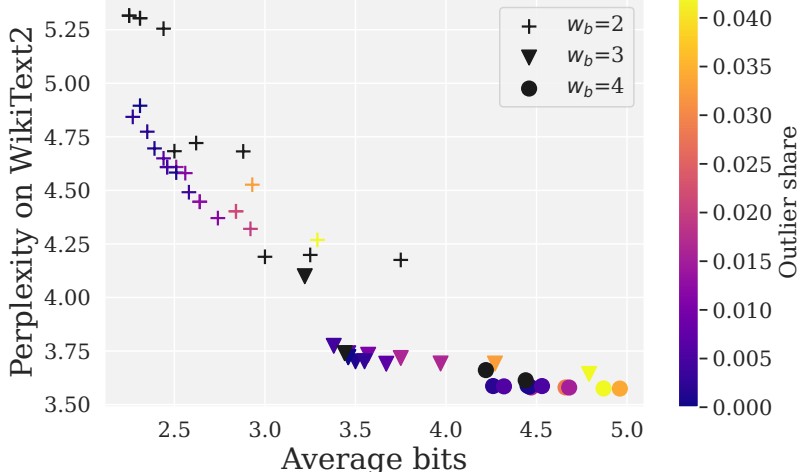

Figure 8: Perplexity of WikiText2 vs average number of bits. Different markers denote different $b_w$. Black colors correspond to quantization configurations without outliers and the brightness of the color is proportional to the outlier rate.

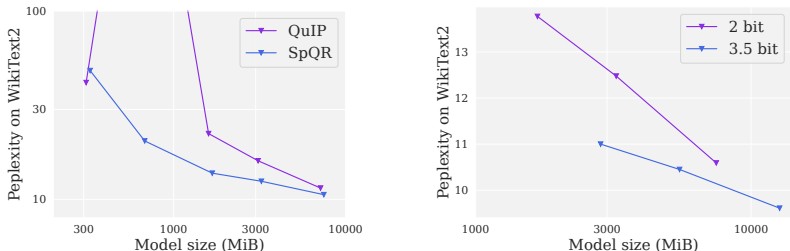

Figure 9: (**left**) SpQR vs QuIP on OPT-family. Smallest models (OPT-125m, OPT-350m) are not depicted since both methods give perplexity $> 100$. (**right**) 2-bit SpQR vs 3.5-bit SpQR QuIP on OPT-family.

approximately equal budget SpQR outperforms AWQ and performs on par with SqueezeLLM at $4$ bit, while being better at $3.25$. We point out that our method is more memory-efficient and scalable than SqueezeLLM since the latter approach requires computing gradients and backpropagating, whereas SpQR needs only forward passes and can be executed in a block-per-block manner without the need to load whole model on GPU. One get most of the advantage from SpQR at bitwidths in between two integer values, for instance, 3.5 bits, where one can utilize small groups and outliers simultaneously.

# H SPQR ALGORITHM WITH AUXILIARY FUNCTIONS

Algorithm 2 provides an extended definition for Algorithm 1 from the main paper with auxiliary functions for fitting and applying min-max quantization.

**Algorithm 2** SpQR quantization algorithm: the left snippet describes the full procedure, the right side contains subroutines for min-max quantization, bilevel quantization and finding outliers.

```
func SPQRQUANTIZE(W, X, b, β₁, β₂, τ, λ)
```
**Input:** $W \in \mathcal{R}^{m \times n}$ — weight matrix,
$X \in \mathcal{R}^{n \times d}$ — calibration data,
$b$ — the base number of quantization bits,
$\beta_1, \beta_2$ — quantization group sizes,
$\tau$ — sensitivity outlier threshold
$\lambda$ — hessian regularizer,

1: $E :=$ float_matrix$(m, n)$    // L2 error
2: $H := 2XX^T$    // L2 error hessian, $\mathcal{R}^{n \times n}$
3: $H^{\text{ic}} :=$ Cholesky$((H + \lambda \mathbf{I})^{-1})$
4: $Q :=$ int_matrix$(m, n)$    // quantized weight
5: $\mathcal{O} := \emptyset$    // a set of all outliers
6: $\mathcal{S} := \emptyset$    // a set of quantization statistics
7: **for** $i = 1, \beta_1, 2\beta_1, \ldots n$ **do**
8:     $W_{:,i:i+\beta_1}, \mathcal{O} :=$ outliers$(W_{:,i:i+\beta_1}, H^{\text{ic}}_{i:(i+\beta_1),i:(i+\beta_1)}\mathcal{O})$
9:     $\hat{s}, \hat{z}, \mathcal{S} :=$ fit_statistics$(W_{:,i:i+\beta_1}, \mathcal{S}, \mathcal{O})$
10:     **for** $j = i, \ldots, i + \beta_1$ **do**
11:       $Q_{:,j} :=$ quantize$(W_{:,j}, \hat{s}, \hat{z})$
12:       $\vec{w}_q :=$ dequantize$(Q_{:,j}, \hat{s}, \hat{z})$
13:       $E_{:,j} := (W_{:,j} - \vec{w}_q)/H^{\text{ic}}_{j,j} \cdot (1 - \text{is\_outlier}(W_{:,j}, \mathcal{O}))$
14:       $W_{:,j:(i+\beta_1)} := W_{:,j:(i+\beta_1)} - E \cdot H^{\text{ic}}_{j,j:(i+\beta_1)}$
15:     $W_{:,(i+\beta_1):n} := W_{:,(i+\beta_1):n} - E \cdot H^{\text{ic}}_{i:(i+\beta_1),i:(i+\beta_1)}$
16: $S_q, Z_q, S_s, Z_s, S_z, Z_z :=$ gather_statistics$(\mathcal{S})$
17: $W_{sparse} =$ gather_outlier_matrix$(W, \mathcal{O})$
18: **return** $Q, S_q, Z_q, S_s, Z_s, S_z, Z_z, W_{sparse}$

```
func quantize(M, s⃗, z⃗)
```
1: **return** $\lfloor M/\vec{s} + \vec{z} + 0.5 \rfloor$

```
func dequantize(Q, s⃗, z⃗)
```
1: **return** $\vec{s} \cdot (Q - \vec{z})$

```
func fit_quantizer(M, β)
```
1: $\vec{m} :=$ flatten$(M)$
2: $\vec{s}, \vec{z} :=$ vectors$()$
3: **for** $i = 1, \beta_1, 2\beta_1, \ldots \dim(m)$ **do**
4:     $s_i := \frac{\max(\vec{m}_{i:i+\beta}) - \min(\vec{m}_{i:i+\beta})}{2^b - 1}$
5:     $z_i := -\min(\vec{m}_{i:i+\beta})/s_i$
6: **return** $\vec{s}, \vec{z}$

```
func error(W, Hⁱᶜ)
```
1: $\vec{s}, \vec{z} :=$ fit_quantizer$(W, \beta_1)$
2: $W_q :=$ quantize$(W, \vec{s}, \vec{z})$
3: $E := (W - W_q)/H^{\text{ic}}$
4: **return** $E^2$

```
func outliers(W, Hⁱᶜ, O)
```
1: $E_{\text{base}} =$ error$(W, H^{\text{ic}})$
2: **for** $i = 1, \ldots, \beta_1$ **do**
3:     $loo := \{1, 2, ..., \beta_1\}/\{i\}$
4:     $E_{\text{ol}} =$ error$(W_{:,loo}, H^{\text{ic}}_{\text{loo,loo}})$
5:     $I_o =$ select$(E_{\text{base}} - E_{\text{ol}} > \tau)$
6:     $\mathcal{O} := \mathcal{O} \cup I_o$
7: **return** $W, \mathcal{O}$

```
func fit_statistics(W, S, O)
```
1: $W := W \cdot (1 - \text{is\_outlier}(W, O))$
2: $\vec{s}, \vec{z} :=$ fit_quantizer$(W, \beta_1)$
3: // $\vec{s}$ for scales, $\vec{z}$ for zero points
4: $\vec{s}_s, \vec{z}_s :=$ fit_quantizer$(\vec{s}, \beta_2)$
5: $\vec{s}_z, \vec{z}_z :=$ fit_quantizer$(\vec{z}, \beta_2)$
6: $\vec{s}_q :=$ quantize$(\vec{s}, \vec{s}_s, \vec{z}_s)$
7: $\vec{z}_q :=$ quantize$(\vec{z}, \vec{s}_z, \vec{z}_z)$
8: $\mathcal{S} := \mathcal{S} \cup \{s_q, s_s, s_z, z_q, s_z, z_z\}$
9: $\hat{s} :=$ dequantize$(s_q, s_s, s_z)$
10: $\hat{z} :=$ dequantize$(z_q, z_s, z_z)$
11: **return** $\hat{s}, \hat{z}, \mathcal{S}$

# I   ADDITIONAL RESULTS FOR LLM COMPRESSION

In this section we report additional quantization results for OPT, LLaMA and Falcon. The OPT results are in Table 7 on WikiText2, Penn Treebank, and C4 datasets. Table 5 reports perplexities for the Llama 2 model with both 4-bit and near-lossless (4.7-bit) compression.

Next, we report results for LM eval harness for LLaMA (Table 8), as well as for Falcon-7B and Falcon-40B models (Table 9).

In addition, we report results for GSM-8K (Table 10) and Humaneval (Table 11) benchmarks.

**Llama 2**

| Size | Method | Avg bits | Wiki2 | C4 | PTB |
|------|--------|----------|-------|------|------|
| 7B | – | 16.00 | 5.47 | 6.97 | 20.82 |
| | SpQR | 4.63 | 5.53 | 7.03 | 21.20 |
| | GPTQ | 4 | 5.84 | 7.41 | 55.78 |
| | SpQR | 3.98 | 5.65 | 7.19 | 21.95 |
| 13B | – | 16.00 | 4.57 | 6.05 | 30.99 |
| | SpQR | 4.71 | 4.62 | 6.10 | 32.45 |
| | GPTQ | 4 | 4.86 | 6.40 | 33.54 |
| | SpQR | 3.98 | 4.70 | 6.21 | 31.84 |
| 70B | – | 16.00 | 3.12 | 4.97 | 14.87 |
| | SpQR | 4.70 | 3.16 | 5.00 | 15.01 |
| | GPTQ | 4 | 3.36 | 5.17 | 16.08 |
| | SpQR | 3.97 | 3.25 | 5.07 | 15.06 |

Table 5: Perplexity on WikiText2 (Merity et al., 2016), C4 (Raffel et al., 2020) and Penn Treebank (Marcus et al., 1994) for SpQR and GPTQ baselines with Llama 2. We can see that SpQR reaches performances within 1% of the perplexity with less than 4.71 bits per parameter. We also see that for 4-bits per parameter SpQR significantly improves on GPTQ.

**LLaMA**

| Size | Method | Avg bits | Wiki2 | C4 |
|------|--------|----------|-------|------|
| 7B | – | 16.00 | 5.68 | 7.08 |
| | SqueezeLLM | 3.24 | 6.13 | 7.56 |
| | SpQR | 3.24 | 6.01 | 7.45 |
| | AWQ | 4.12 | 5.87 | 7.29 |
| | SqueezeLLM | 4.07 | 5.79 | 7.20 |
| | SpQR | 4.08 | 5.79 | 7.19 |
| 13B | – | 16.00 | 5.09 | 6.61 |
| | SqueezeLLM | 3.24 | 5.45 | 6.92 |
| | SpQR | 3.24 | 5.33 | 6.84 |
| | AWQ | 4.12 | 5.2 | 6.72 |
| | SqueezeLLM | 4.07 | 5.17 | 6.69 |
| | SpQR | 4.08 | 5.17 | 6.68 |
| 30B | – | 16.00 | 4.10 | 5.98 |
| | SqueezeLLM | 3.25 | 4.44 | 6.23 |
| | SpQR | 3.24 | 4.39 | 6.18 |
| | AWQ | 4.12 | 4.24 | 6.07 |
| | SqueezeLLM | 4.07 | 4.20 | 6.05 |
| | SpQR | 4.08 | 4.19 | 6.04 |

Table 6: Perplexity on WikiText2 (Merity et al., 2016), C4 (Raffel et al., 2020) for SpQR, AWQ and SqueezeLLM quantization of LLaMA.

**OPT**

| Size | Method | Avg bits | Wiki2 | C4 | PTB | Size | Method | Avg bits | Wiki2 | C4 | PTB |
|------|--------|----------|-------|------|------|------|--------|----------|-------|------|------|
| 6.7B | – | 16.00 | 10.86 | 11.74 | 13.09 | 30B | – | 16.00 | 9.56 | 10.69 | 11.84 |
| | SpQR | 4.27 | 10.81 | 11.88 | 13.17 | | SpQR | 4.26 | 9.50 | 10.73 | 11.88 |
| | RTN | 4 | 12.10 | 13.38 | 16.09 | | RTN | 4 | 10.97 | 11.90 | 14.17 |
| | GPTQ | 4 | 11.39 | 12.15 | 13.80 | | GPTQ | 4 | 9.63 | 10.80 | 11.98 |
| | SpQR | 3.94 | 11.04 | 11.98 | 13.33 | | SpQR | 3.94 | 9.54 | 10.78 | 11.93 |
| 13B | – | 16.00 | 10.12 | 11.20 | 12.34 | 66B | – | 16.00 | 9.33 | 10.28 | 11.36 |
| | SpQR | 4.27 | 10.22 | 11.27 | 12.41 | | SpQR | 4.23 | 9.37 | 10.32 | 11.40 |
| | RTN | 4 | 11.32 | 12.35 | 15.4 | | RTN | 4 | 110 | 249 | 274 |
| | GPTQ | 4 | 10.31 | 11.36 | 12.58 | | GPTQ | 4 | 9.55 | 10.50 | 11.58 |
| | SpQR | 3.93 | 10.28 | 11.34 | 12.52 | | SpQR | 3.91 | 9.32 | 10.35 | 11.43 |

Table 7: Perplexity on WikiText2 (Merity et al., 2016), C4 (Raffel et al., 2020) and Penn Treebank (Marcus et al., 1994) for SpQR and round-to-nearest (RTN) and GPTQ baselines with OPT. We can see that SpQR reaches performances within 1% of the perplexity with less than 4.3 bits per parameter. We also see that for 4-bits per parameter SpQR significantly improves on GPTQ with an improvement as large as the improvement from RTN to GPTQ.

**LLaMA**

| Size | Method | Avg bits | Winogrande | Piqa | Hellaswag | Arc easy | Arc challenge | Avg score |
|------|--------|----------|------------|------|-----------|----------|---------------|-----------|
| 7B | – | 16.00 | 67.09 | 78.32 | 56.41 | 67.38 | 38.23 | 61.492 |
|  | SpQR | 4.63 | 67.48 | 78.45 | 56.01 | 67.13 | 38.23 | 61.460 |
|  | RTN | 4 | 64.72 | 76.44 | 53.49 | 63.51 | 36.60 | 58.952 |
|  | GPTQ | 4 | 65.35 | 77.58 | 54.99 | 63.55 | 36.35 | 59.564 |
|  | SpQR | 3.45 | 67.48 | 78.13 | 55.27 | 65.87 | 38.05 | 60.960 |
| 13B | – | 16.00 | 70.09 | 78.89 | 59.11 | 74.54 | 43.94 | 65.314 |
|  | SpQR | 4.63 | 69.77 | 78.94 | 59.02 | 74.37 | 43.17 | 65.054 |
|  | RTN | 4 | 69.61 | 78.24 | 57.34 | 72.56 | 42.58 | 64.066 |
|  | GPTQ | 4 | 69.06 | 78.40 | 58.04 | 73.23 | 43.26 | 64.398 |
|  | SpQR | 3.45 | 68.90 | 78.73 | 58.22 | 73.27 | 42.75 | 64.374 |
| 30B | – | 16.00 | 72.93 | 80.96 | 62.66 | 75.34 | 46.76 | 67.730 |
|  | SpQR | 4.69 | 72.93 | 81.01 | 62.50 | 76.05 | 47.18 | 67.934 |
|  | RTN | 4 | 72.06 | 79.05 | 60.61 | 70.66 | 42.24 | 64.924 |
|  | GPTQ | 4 | 72.61 | 79.92 | 61.07 | 71.8 | 44.28 | 65.936 |
|  | SpQR | 3.49 | 73.32 | 80.47 | 61.96 | 74.75 | 46.93 | 67.486 |
| 65B | – | 16.00 | 77.43 | 81.50 | 63.95 | 75.17 | 47.10 | 69.030 |
|  | SpQR | 4.71 | 76.95 | 81.56 | 63.76 | 75.25 | 46.93 | 68.890 |
|  | RTN | 4 | 75.14 | 81.45 | 62.79 | 72.64 | 44.97 | 67.398 |
|  | GPTQ | 4 | 75.85 | 80.79 | 62.91 | 74.20 | 46.59 | 68.068 |
|  | SpQR | 3.52 | 76.09 | 81.18 | 63.54 | 74.37 | 45.05 | 68.046 |

Table 8: LM eval harness results on LLaMA models.

**Falcon**

| Size | Method | Avg bits | Winogrande | Piqa | Hellaswag | Arc easy | Arc challenge | Avg score |
|------|--------|----------|------------|------|-----------|----------|---------------|-----------|
| 7B | – | 16.00 | 67.32 | 79.49 | 57.77 | 74.71 | 40.1 0 | 63.878 |
|  | SpQR | 4.44 | 67.09 | 79.16 | 57.21 | 73.86 | 38.99 | 63.262 |
|  | RTN | 4.00 | 65.51 | 77.37 | 51.86 | 68.69 | 33.7 | 59.426 |
|  | GPTQ | 4.00 | 66.38 | 79.11 | 56.68 | 73.15 | 38.48 | 62.760 |
|  | SpQR | 3.49 | 67.88 | 79.54 | 57.08 | 74.03 | 39.08 | 63.522 |
| 40B | – | 16.00 | 76.62 | 82.32 | 64.06 | 82.03 | 50.26 | 71.058 |
|  | SpQR | 4.46 | 76.48 | 82.1 | 63.8 | 81.78 | 50.77 | 70.986 |
|  | RTN | 4.00 | 75.69 | 80.30 | 60.52 | 79.92 | 49.83 | 69.252 |
|  | GPTQ | 4.00 | 75.93 | 81.23 | 63.05 | 80.85 | 50.00 | 70.212 |
|  | SpQR | 3.45 | 76.32 | 81.77 | 63.70 | 81.10 | 49.83 | 70.544 |

Table 9: LM eval harness results on Falcon models.

**GSM-8k**

| Model | Method | Avg bits | Accuracy (%) |
|-------|--------|----------|--------------|
| MetaMath-7B | None | 16 | 66.9 |
|  | RTN | 4 | NaN |
|  | GPTQ | 4 | 64.7 |
|  | SpQR | 3.98 | 66.5 |

**HumanEval**

| Model | Method | Avg bits | Pass@1 | Pass@10 |
|-------|--------|----------|--------|---------|
| CodeLLama-7b | None | 16 | 40.0 | 58.0 |
|  | RTN | 4 | NaN | NaN |
|  | GPTQ | 4 | 32.1 | 52.2 |
|  | SpQR | 3.98 | 36.3 | 54.6 |

Table 10: GSM-8k results including MetaMath-7B model performances with various quantization methods.

Table 11: HumanEval results for CodeLLama-7b with different quantization techniques.

## J   COMPARISON ACROSS QUANTIZATION METHODS AT THE SAME ACCURACY LEVEL

In addition to comparison at the same bitwidth we perform comparison at the same accuracy level. The Table12 shows that GPTQ requires 0.5-0.6 more bits per parameter to match the performance of SpQR configuration with 3.5 bits per parameters on average, and 0.3-0.4 to match lossless compression configuration. This difference can be critical in some cases. For example, with 3.5 bit per parameter one can fit Llama-2-70b on a single V100 with 32Gb and have some space for KV cache, which would be impossible for GPTQ quantization with the same accuracy without weight offloading.

**Model Performance**

| Size | Method | Avg bits | WikiText2 | C4 |
|---|---|---|---|---|
| 7B | GPTQ | 4.07 | 5.88 | 7.28 |
| | SpQR | 3.45 | 5.88 | 7.33 |
| | GPTQ | 5.07 | 5.72 | 7.12 |
| | SpQR | 4.63 | 5.73 | 7.13 |
| 13B | GPTQ | 4.04 | 5.26 | 6.76 |
| | SpQR | 3.45 | 5.25 | 6.77 |
| | GPTQ | 5.04 | 5.13 | 6.64 |
| | SpQR | 4.63 | 5.13 | 6.64 |
| 30B | GPTQ | 4.02 | 4.28 | 6.11 |
| | SpQR | 3.49 | 4.29 | 6.11 |
| | GPTQ | 5.01 | 4.14 | 6.01 |
| | SpQR | 4.69 | 4.14 | 6.01 |
| 65B | GPTQ | 4.01 | 3.71 | 5.74 |
| | SpQR | 3.52 | 3.71 | 5.73 |
| | GPTQ | 5.01 | 3.58 | 5.65 |
| | SpQR | 4.71 | 3.57 | 5.64 |

Table 12: Performance comparison on WikiText2 and C4 datasets for GPTQ and SpQR methods across different model sizes and weight bits (wbits).

## K  CHOICE OF OPTIMAL LLM CONFIGURATION FOR SPECIFIC HARDWARE

In the preceding discussion, we were searching for optimal model configuration given some compression target without targeting any specific hardware or device. However, the question practitioner willing to deploy a model for a specific application would ask is: What is the best model and compression setup for a given memory constraint?

In this section, we provide a list of recommendations for the choice of the best LLaMA model and the corresponding compression level that fits into the device memory (RAM or VRAM) without the need of offloading model parameters and activations. We cover a range of available budgets from mobile devices to high-end workstation GPUs. Recommendations are presented in Table 13.

Table 13: Choice of the best LLaMA for a given memory constraint.

| Device | Memory (GiB) | LLaMA | $\bar{b}$ |
|---|---|---|---|
| iPhone13 | 4 | 7B | $\leq 3.5$ |
| iPhone14 | 6 | 7B | $\simeq 4.5$ |
| | | 13B | $\leq 3.5$ |
| Consumer laptop | 8 | 13B | $\leq 4$ |
| RTX4070 | 10-12 | 14B | $\simeq 4.5$ |
| RTX4080 | 16 | 30B | $\leq 4$ |
| RTX4090 | 24 | 30B | $\simeq 4.5$ |
| V100 | 32 | 65B | $\leq 3.5$ |
| A6000 | 48 | 65B | $\simeq 4.5$ |

## L    SENSITIVITY TO RANDOM SEED

The experiments we report throughout Section 5 use one fixed random seed (the default value from the supplementary code). To verify that our results are robust to randomness, we run SpQR with 5 random seeds (0-5) and measure the adjusted standard deviation.

For this evaluation, we compress LLaMA-65B with SpQR using $b_w = b_z = b_s = 3$ and $\beta_1 = \beta_2 = 16$, which corresponds to 3.625 bits per parameter. The resulting perplexity scores are $3.75 \pm 0.003$ (WikiText2), $7.03 \pm 0.01$ (Penn Treebank) and $5.75 \pm 0.00086$ (C4). In addition to the chosen random seed, these standard deviations can be affected by the inherent nondeterminism of GPU computation. Overall, the standard deviations are at least one order of magnitude smaller than the difference between SpQR, GPTQ, and RTN.

## M    GENERATIVE EXAMPLES

Finally, we showcase several examples of how SpQR quantization affects the generated samples. For this evaluation, we take several prompts and use the compressed language model to continue generating text from these prompts. We compare the original LLaMA-65B and two quantized versions: SpQR and RTN-4bit. More specifically, we use the SpQR configuration that corresponds to near-lossless compression from Table 1. We use greedy autoregressive inference for all generated samples to ensure reproducibility. The examples in Figure 10 show that all models produce a valid text, but SpQR matches the 16-bit model more frequently. The near-lossless algorithm also seems to produce more semantically similar texts.

## N    BROADER IMPACT

Our method enables the deployment of high-quality LLMs in the 7-13B parameters range to memory-limited devices such as laptops and phones. With our method, it is possible to develop specialized 7B LLMs in hassle-free 16-bit and then enable the deployment of such LLMs to phones by applying SpQR. Since SpQR is practically lossless, this ensures a reliable performance level for deployed LLMs which is important for consumer applications. Since mobile phones are ubiquitous and LLMs powerful general-purpose tools, SpQR might have a wide-reaching effect on how LLMs are used by the general population to complete useful tasks.

LLMs are inherently a dual-use technology that can bring both significant benefits and serious harm. The ethical and societal risks of LLMs range from deliberate malicious use (e.g. generating spam) and accidental misuse to adverse economic side-effects (Weidinger et al., 2021). However, we believe that the marginal impact of SpQR will be positive or neutral since the LLMs we use are already openly available. Better quantization algorithms like SpQR let users with low-end devices run larger and generally more accurate language models. In other words, our algorithm does not create models with new capabilities (and risks): it only makes existing models more accessible.

| Prompt: | Every time data scientist Kim ran her new algorithm, it kept outputting the same unexpected result: a recipe for |
|---|---|
| 16bit: | iced tea. Kim was working on a project to predict the likelihood of a patient being readmitted to the hospital within 30 days of discharge. She was using a machine learning algorithm to analyze the data, and the algorithm |
| SpQR: | iced tea. Kim was working on a project to predict the likelihood of a patient being readmitted to the hospital within 30 days of discharge. She had been given a dataset of 100,000 |
| RTN 4-bit: | iced tea. \I'm not sure what's going on," she said. \I've been running it for days, and it keeps giving me the same result." \I'm not sure what's going on |
| Prompt: | Data scientist Grace was working late one night when she discovered a glitch in the matrix of her data. It spelled out |
| 16bit: | 100% proof that the world was going to end in 10 days. She tried to tell her boss, but he didn't believe her. She tried to tell her friends, but they didn't believe her |
| SpQR: | 100% proof that the world was going to end in 10 days. She tried to tell her boss, but he just laughed and said, \I'm not going to believe in some data glitch." She |
| RTN 4-bit: | 100% in binary code. She was so excited she ran to tell her boss, but he was busy. She ran to tell her colleague, but he was busy. She ran to tell her friend, but she |
| Prompt: | Last Tuesday, just as the sun was setting, my neighbor's dog, Rover, started to speak to me in perfect French. He said |
| 16bit: | "Je suis un chien." I was shocked. I said, "Moi aussi." I'm not sure what to make of this. I'm not sure what to make of anything. I'm not sure what |
| SpQR: | "Je suis un chien." I was shocked. I said, "Rover, you can speak French?" He said, "Oui, je parle francais." I said, "How did you learn to speak French?" |
| RTN 4-bit: | "Bonjour, Monsieur. Je m'appelle Rover. Je suis un chien. Je suis un chien français. Je suis un chien français qui parle français. Je suis un chien français qui parle français |
| Prompt: | In a parallel universe where cats are the rulers of Earth, our feline king was pondering his first decree |
| 16bit: | " to make all dogs illegal. "I"m not sure that"s a good idea," said his chief adviser, a cat named Whiskers. "Why not?" asked the king. "Because, |
| SpQR: | " to make all dogs illegal. "I"m not sure that"s a good idea," said his chief adviser, a cat named Whiskers. "Why not?" asked the king. "Because, |
| RTN 4-bit: | " to make the world a better place for cats. He was about to sign the decree when he was interrupted by a knock on the door. "Come in," he said. The door opened and a cat entered. |

Figure 10: Texts generated by different quantized LLaMA-65B models with the same prompt.

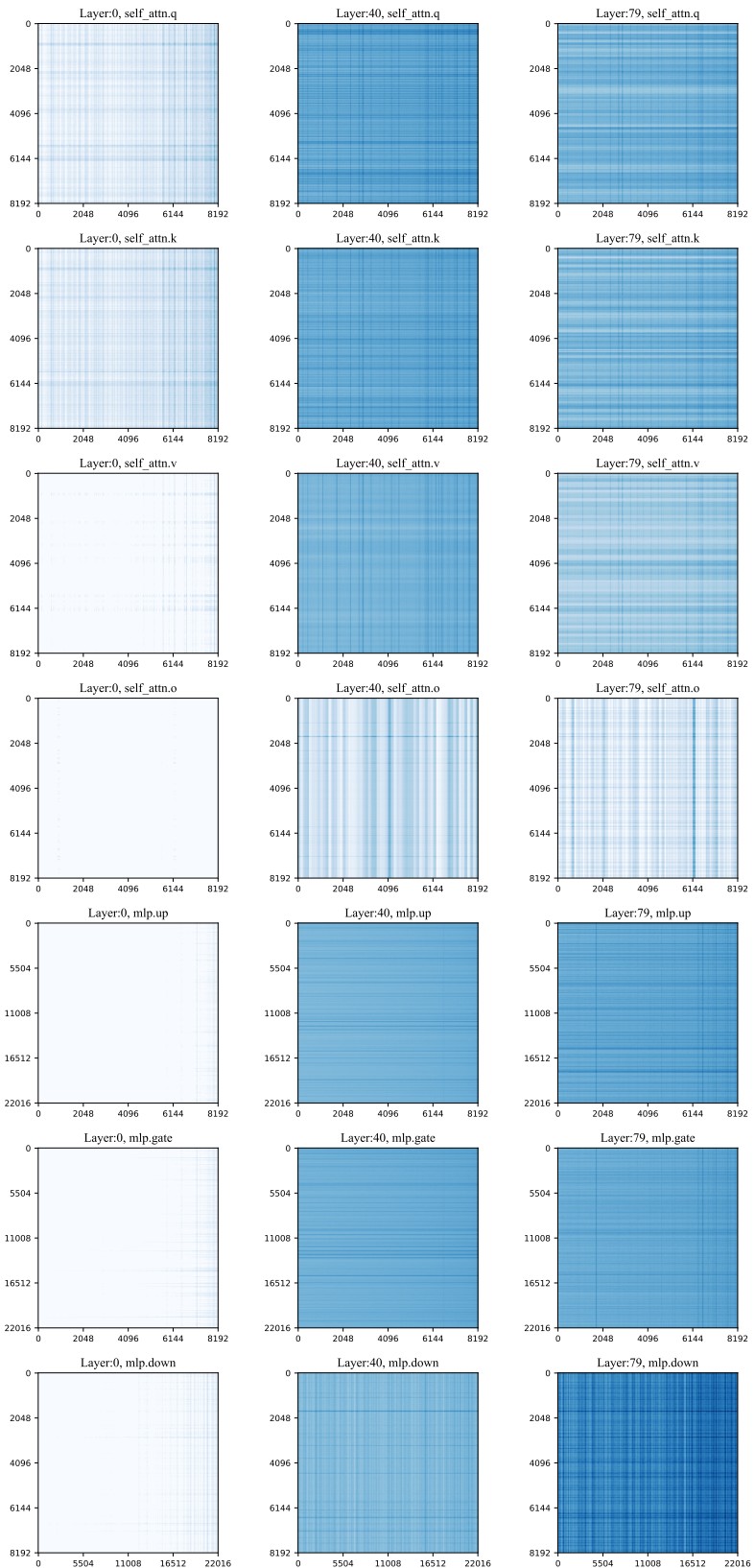

Figure 11: A grid of weight log-sensitivities for LLaMA-65B for 3-bit GPTQ compression with per-row quantization statistics. Each row corresponds to a specific layer type (e.g. attention query, mlp gate), and the columns represent layer depth.

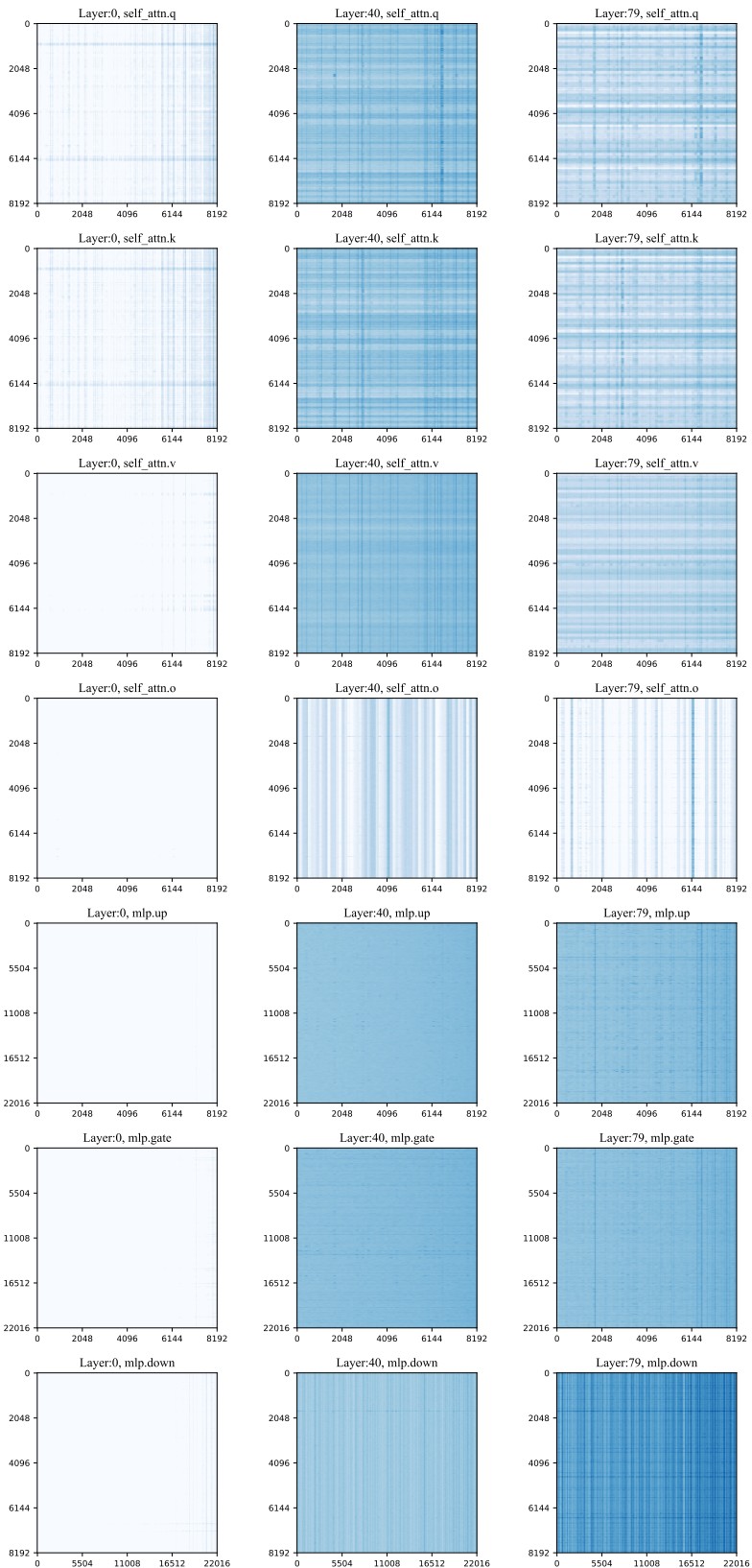

Figure 12: A grid of weight log-sensitivities for LLaMA-65B for 3-bit GPTQ compression with group-wise quantization of block size 128. Each row corresponds to a specific layer type (e.g. attention query, mlp gate), and the columns represent layer depth.

