# OpenReview forum: "SpQR: A Sparse-Quantized Representation for Near-Lossless LLM Weight Compression"
_ICLR.cc/2024/Conference — ICLR 2024 poster_

### Official Review · Reviewer_54fV · 2023-10-30

**Soundness:** 2 fair
**Presentation:** 3 good
**Contribution:** 3 good
**Rating:** 6
**Confidence:** 4

**Summary:**

Through extensive analysis of large language model outliers, the proposed method, SpQR, adopts a dual-stage weight-only quantization with small group sizes, achieving 3 or 4-bit precision per parameter. For the outliers, SpQR preserves them in FP16 with an unstructured format using an indices matrix. SpQR provides details on the implementation of both encoding and decoding algorithms. Experimental results show competitive performance on next-token generation tasks in language modeling, such as Wikitext2, C4, PTB with Falcon, and LLaMA. Notably, the provided GPU inference algorithm exhibits faster latency compared to 16-bit baselines while maintaining similar accuracy.

**Strengths:**

* Detailed analytical demonstrations of outliers in large language models.
* Offers a detailed breakdown of how the two-stage quantization works in conjunction with the sparse representation for outliers.
* Sets aside outlier values in a sparse, unstructured format to keep the accuracy of the language model's performance, which is a more advanced concept than previous methods.
* The overall problem statement and its algorithmic solution are clear and easy to understand.

**Weaknesses:**

1. *Evaluation Benchmarks:* While the authors have acknowledged this in their limitations, the absence of performance evaluations on trendy benchmarks for generative models (such as MMLU, mathematical reasoning performance like GSM8K, or coding abilities such as humaneval) represents a weakness. Evaluating at least one of these benchmarks is crucial to better ascertain the proposed algorithm's effectiveness.
2. *Quantization Baselines:* SpQR demonstrates good accuracy using a very small group size. Therefore, I kindly suggest comparing it with baselines like RTN, GPTQ [1] and AWQ [2] by applying a group size. While it's important to compare accuracy at identical bit-width levels, it's equally crucial to evaluate inference latency at the same accuracy level. In this context, SpQR might offer reduced memory usage. Since SpQR introduces a new representation by combining dual quantization with sparse outliers, it's essential to provide a presentation that underscores the advantages of SpQR in relation to conventional methodologies.
3. *Inference Time:* SpQR employs a two-stage quantization process to utilize a very small group size, allowing for fine-grained representation and enhancing accuracy through the use of sparse representation for outliers. While it is evident that this proposed method will yield very positive results in terms of accuracy, there is a growing concern that the associated latency overhead needs to be better managed. In other words, the method presents a well-analyzed problem statement combined with easily understandable techniques that have high accuracy. However, it seems to have a weakness in the latency overhead of the two-stage quantization with such a small group size and the latency overhead for the sparse representation of outliers. Therefore, it would have been more beneficial if there had been a more varied presentation of the latency results, which arise from combining sparse matrix multiplication with quantization matrix multiplication. Specifically, in the "Inference Time" section and paragraph, the speed of the end-to-end proposed algorithm is measured. However, it would have been a better presentation if the latencies without sparse outliers and with sparse outliers were both disclosed, allowing readers to recognize the performance trade-off easily. To delve into more specifics, when sparse outliers are excluded, there should be a new baseline (such as existing prior kernels [1, 2] for weight-only quantization) comparing the latency of the quantized model where the proposed two-stage quantization has been applied. Furthermore, discussing the additional overhead when introducing sparse outliers after such a comparison would have been an excellent addition.

Ideally, to assess the performance of SpQR, it would enhance clarity if the author could specify the difference in accuracy when comparing the proposed SpQR to 4-bit group-wise methods like RTN, GPTQ [1], and AWQ [2]. At the same time, understanding the inference latency of SpQR, especially given its very small group size in comparison to these 4-bit group-wise weight-only quantization methods, would be beneficial. Lastly, highlighting its memory efficiency would further improve the clarity.

[1] Frantar, Elias, et al. "OPTQ: Accurate quantization for generative pre-trained transformers." The Eleventh International Conference on Learning Representations. 2022.
[2] Lin, Ji, et al. "AWQ: Activation-aware Weight Quantization for LLM Compression and Acceleration." arXiv preprint arXiv:2306.00978 (2023).

**Questions:**

* There are typos in the manuscript (in sections 5, 3rd paragraph 'emory'). I suggest revising them.
* In the "Ablations" paragraph of section 5, the mention of "16-bit statistics" seems to require clarification regarding the group size.

---

> ### Author Response · Authors · 2023-11-18
> **Thank you for your review (part 1/2)**
>
> **Q1: [...] the absence of performance evaluations on trendy benchmarks for generative models (such as MMLU, [...] GSM8K, or[...] humaneval) represents a weakness [...]**
>
> **A1**: To address this concern, we provide MMLU and GSM-8K results below. We believe that perplexity measures are still very important: Previous work [1] showed that perplexity evaluations have a very strong correlation (-0.94) with performance on a diverse set of zeroshot tasks. This means that the perplexity evaluations generalize to many different tasks.
>
> Following your suggestion we evaluate our method with MetaMath-7B, a recent version of Llama-2 specialized on math problem-solving, and  CodeLlama-7b, a popular code generation model. One can see that SpQR is close to fp16 model performance with 4 bit quantization while outperforming GPTQ by ~2\%. Drops on HumanEval dataset are more significant, but SpQR is still much better than GPTQ at the same bitwidth. These results correlate well with perplexity measure differences. We do not report pass@100 score, as it appears to be extremely noisy.  RTN quantization turns out to be numerically unstable and produces models with random performance.
>
> Table 1. GSM-8k results
>
> |	model	| method | wbits | Accuracy (%) |
> |:-----------:|:------:|:-----:|:------------:|
> | MetaMath-7B |  None  |   16  | 	66.9 	|
> |         	|   RTN  |   4   |  	NaN 	|
> |         	|  GPTQ  |   4   | 	64.7 	|
> |         	|  SpQR  |  3.98 | 	66.5 	|
>
> Table 2. HumanEval results
>
> | 	model	| method | wbits | pass@1 | pass@10 |
> |:------------:|:------:|:-----:|:------:|:-------:|
> | CodeLLama-7b |  None  |   16  |  40.0  |   58.0  |
> |          	|   RTN  |   4   |	NaN  |	NaN	|
> |          	|  GPTQ  |   4   |  32.1  |   52.2  |
> |          	|  SpQR  |  3.98 |  36.3  |   54.6  |
>
>
> **Q2: [...] SpQR demonstrates good accuracy using a very small group size. Therefore, I kindly suggest comparing it with baselines like RTN, GPTQ [1] and AWQ [2] by applying a group size. [...] it's equally crucial to evaluate inference latency at the same accuracy level. [...]**
>
> **A2**: The performance of GPTQ is very competitive at a group size of 128 and we were not able to improve performance by much using a smaller group size. This is why we kept running the experimental settings from the main GPTQ experimental setup. We will run these experiments and will provide these results. In terms of latency the overhead of loading groups comes mainly from the memory footprint of the scales and zeropoints and not the group size itself. As such, the average bit-footprint translates almost 1:1 to inference latency.
>
> **Q3: [...] there is a growing concern that the associated latency overhead needs to be better managed. In other words, [... it seems to have a weakness in the latency overhead of the two-stage quantization with such a small group size and the latency overhead for the sparse representation of outliers. Therefore, it would have been more beneficial if there had been a more varied presentation of the latency results [...]**
>
> **A3**: Small group sizes do not affect latency for our implementation. Since every warp of threads on the GPU processes multiple values in the matrix multiplication, we can load multiple groups with each warp of threads. The total values loaded remain the same independent of group size. This means that inference speed is not dependent on the group size but just on the average amount of bits in the representation. The latency is affected by the sparse representations, but with the right implementations we can still generate about 115 tokens per second from a 7B model on a single A100 GPU. As such, our implementation is still fast enough to be practical despite the sparse representations.
>
>
> [1] Dettmers et al, The case for 4-bit precision: k-bit Inference Scaling Laws

---

> ### Author Response · Authors · 2023-11-18
> **response part 2/2**
>
> **Q5: Ideally, to assess the performance of SpQR, it would enhance clarity if the author could specify the difference in accuracy when comparing the proposed SpQR to 4-bit group-wise methods like RTN, GPTQ [1], and AWQ [2].**
>
> **A5**: As suggested, in addition to comparison at the same bitwidth we perform comparison at the same accuracy level. The table below shows that GPTQ requires 0.5-0.6 more bits per parameter to match the performance of SpQR configuration with 3.5 bits per parameters on average, and 0.3-0.4 to match lossless compression configuration. This difference can be critical in some cases.
> For example, with 3.5 bit per parameter one can fit Llama-2-70b on a single V100 with 32Gb
> and have some space for KV cache, which would be impossible for GPTQ quantization with the same accuracy without weight offloading.
>
> | Model | Method | wbits | wikitext2 |  c4  |
> |:-----:|:------:|:-----:|:---------:|:----:|
> |   7   |  GPTQ  |  4.07 |	5.88   | 7.28 |
> |   	|  SpQR  |  3.45 |	5.88   | 7.33 |
> |   	|  GPTQ  |  5.07 |	5.72   | 7.12 |
> |   	|  SpQR  |  4.63 |	5.73   | 7.13 |
> |   13  |  GPTQ  |  4.04 |	5.26   | 6.76 |
> |   	|  SpQR  |  3.45 |	5.25   | 6.77 |
> |   	|  GPTQ  |  5.04 |	5.13   | 6.64 |
> |   	|  SpQR  |  4.63 |	5.13   | 6.64 |
> |   30  |  GPTQ  |  4.02 |	4.28   | 6.11 |
> |   	|  SpQR  |  3.49 |	4.29   | 6.11 |
> |   	|  GPTQ  |  5.01 |	4.14   | 6.01 |
> |   	|  SpQR  |  4.69 |	4.14   | 6.01 |
> |   65  |  GPTQ  |  4.01 |	3.71   | 5.74 |
> |   	|  SpQR  |  3.52 |	3.71   | 5.73 |
> |   	|  GPTQ  |  5.01 |	3.58   | 5.65 |
> |   	|  SpQR  |  4.71 |	3.57   | 5.64 |
>
>
> **Q6: [...] Lastly, highlighting its memory efficiency would further improve the clarity.**
>
> **A6**: We acknowledge this point. The memory efficiency is determined by the average amount of bits which we report. For example, 3.4 bits per parameter for a 70B model would be 70B*(3.4 bits / 8 bits) = 70B*(0.425 bytes) = 29.75 GB.
>
> **Q7: In the "Ablations" paragraph of section 5, the mention of "16-bit statistics" seems to require clarification regarding the group size.**
>
> **A7**: Thank you, we overlooked this detail and will add this information.
>
> [1] Dettmers et al, The case for 4-bit precision: k-bit Inference Scaling Laws

---

> > ### Comment · Reviewer_54fV · 2023-11-23
> >
> > I appreciate the authors' detailed responses and additional experimental results. Given that the problem statement, which is based on an extensive study of outliers in large language models, and its proposed solution are convincing, this paper offers insights into the quantization field. Although some of the answers do not sufficiently address the questions, I believe the additional experimental results, as well as the in-depth discussion, will be valuable for future research in this field. Therefore, I have decided to raise my original rating.

---

### Official Review · Reviewer_Sumn · 2023-10-31

**Soundness:** 3 good
**Presentation:** 3 good
**Contribution:** 3 good
**Rating:** 8
**Confidence:** 5

**Summary:**

This paper proposes a hybrid sparse-quantization framework named Sparse-Quantized Representations (SpQR), which identifies the isolating outlier weights and stores them in higher precision while compressing all other weights to 3-4 bits. To overcome the challenge of poor GPU support, a sparse-matrix multiplication algorithm is proposed. The proposed achieves compression levels comparable to previous methods with less performance degradation.

**Strengths:**

1. While outlier input features have been observed in existing work, this work is the first to demonstrate that similar outliers occur in the weights, for particular output hidden dimensions.

2. This work not only indicate the advantages of weight quantization in terms of memory saving, but design a specific sparse-matrix multiplication algorithm, which demonstrates the advantage of inference time of proposed in Tab. 3.

3. This paper is well-written and organized, and the supplementary material is sufficiently detailed (such Tab. 10).

**Weaknesses:**

1. The experiment results related to large vision model (such SAM/DINOv2) are expected also.

**Questions:**

1. Are there experimental results that also quantify the activation?

---

> ### Author Response · Authors · 2023-11-18
> **Thank you for your review!**
>
> **Q1: The experiment results related to large vision model (such SAM/DINOv2) are expected also.**
>
> **A2**: We believe we have a good range of language models that we evaluate. Adding vision models can help to test the robustness of our approach. On the other hand, vision models are relatively small compared to LLMs and since the main advantage of our method is the memory savings while maintaining inference speed, it would not be very useful for vision models since they are often not limited by memory.
>
> **Q2: Are there experimental results that also quantify the activation?**
>
> **A2**: Currently, we do not do activation quantization. We could use the outlier mask also for the activations which would solve some problems observed in earlier work. However, we quantizing the activations to 4-bit or less would be a new and very difficult problem. We think outliers + 8-bit inputs with SpQR quantized weights could work well, but we have not run these experiments.

---

> ### Comment · Reviewer_Sumn · 2023-11-23
>
> Thanks for the authors' response. I keep my rating that recommends accept.

---

### Official Review · Reviewer_DynX · 2023-10-31

**Soundness:** 3 good
**Presentation:** 3 good
**Contribution:** 3 good
**Rating:** 6
**Confidence:** 5

**Summary:**

This paper proposes a new quantization method, SpQR, for the weights of LLMs. This method first isolates the outlier weights and keeps them in high precision, then it employs a grouped quantization for higher accuracy. During inference, SpQR uses sparse-matrix multiplication for the outliers and uses dense-quantized matrix multiplication for “base” weights. As a result, compared with the 16-bit model, it can achieve 3.4x compression and 20-30% speedup without any degradation in perplexity.

**Strengths:**

* Give a deep analysis of the outliers in the weights and show different distribution patterns of them, which can help isolate the outliers from the weights.
* Propose a grouped quantization, which is more fine-grained than prior methods, for better model accuracy. Also, propose a two-level quantization to reduce the overhead of the group-wise statistics.
* Experiments show the real speedup on GPUs to demonstrate the effectiveness of the proposed method.

**Weaknesses:**

* The proposed method has limitations and may not be easily applicable to activation quantization in LLM. During inference, the memory accesses for activations also take a large amount of time.
* The unstructured sparse pattern of the outliers is not very friendly to GPUs, and this will impose significant overhead on the computation.

**Questions:**

* In section 4.1, high-sensitivity outliers, there are two types of outliers, i.e., group-wise outliers and individual outliers. In section 4.2, the weight outliers are stored with the Run-Length Coding, which saves the relative index. Are the group-wise outliers used in the proposed method? Or only save individual outliers? For the two types of outliers, do they both use the same format to store?
* As for the weights in multi-head attention and the MLP, is there any difference when quantizing those weights? Or they are all the same?
* Did you compare the model accuracy and the real speedup with the 8-bit weight quantization methods? I just wanna if it is worth quantizing the weights to 4-bit with such a large overhead from the unstructured sparse outliers.
* Can you compare the performance between the quantized GEMM with 4-bit SpQR and the quantized GEMM with a normal 4-bit weight quantization method? It is better to do such ablation to show the performance overhead of the grouped quantization and the unstructured sparse outliers.

---

> ### Author Response · Authors · 2023-11-18
> **Thank you for your review!**
>
> **Q1: The proposed method has limitations and may not be easily applicable to activation quantization in LLM. During inference, the memory accesses for activations also take a large amount of time.**
>
> **A1**: If the batchsize is 1, for example LLM use for a single user, the activations for Llama 2 7B are 4000x smaller than the weight matrix. So for this case, the activation just take 0.025% of overhead. For larger batch sizes, the activations are a larger overhead, but even for batch sizes common in deployments, like ChatGPT, loading the activations for a 7B model would only account for at most about 7% of overhead. As such, this limitation is not a big one.
>
> **Q2: The unstructured sparse pattern of the outliers is not very friendly to GPUs, and this will impose significant overhead on the computation.**
>
> **A2**: We measured the overhead and still provide faster inference than a unquantized model. So yes, there is an overhead, but inference is still fast. With the right implementation, for a 7B model, we could generate about 115 tokens per second on a single RTX 4090 consumer GPU. As such, this would still be fast enough to be practical. Thus, there is no significant practical disadvantage to having the sparse structure.
>
> **Q3: [...] weight outliers are stored with the Run-Length Coding [...] group-wise outliers used in the proposed method? Or only save individual outliers? For the two types of outliers, do they both use the same format to store?**
>
> **A3**: We use the run-length coding for all outliers. This means both the grouped outliers and sparse ones identified in our analysis are quantized in the same way. The group-wise outliers only make up about 0.1 to 0.2% of total weights. As such, having a separate encoding just for these outliers is not very beneficial since the efficiency gains would be minimal.
>
> **Q4: As for the weights in multi-head attention and the MLP, is there any difference when quantizing those weights? Or they are all the same?**
>
> **A4**: We did not study this question in this work, but we studied this before (unpublished results) and we found that essentially the performance per weights is approximately the same. Since MLP layers have in total more weights than all attention layers, this means that MLP layers as a whole have slightly more importance. But for individual parameters in the weights, the importance is roughly the same. As such, there is no major difference between quantizing multi-head attention and MLP layer weights. We see some difference in outliers distribution, but these are small and only account for less than 1% of all quantized values.
>
> **Q5: Did you compare the model accuracy and the real speedup with the 8-bit weight quantization methods? I just wanna if it is worth quantizing the weights to 4-bit with such a large overhead from the unstructured sparse outliers.**
>
> **A5**: Our method should be about 2x slower than 8-bit methods, but reduces memory by more than 2x times. As such, this method is best used for very large models that are difficult to use. For example, our method is ideal for a 7B model on mobile devices, or a 13B model on laptops, or a 65B model on consumer GPUs. 8-bit methods are more suitable when one runs large batch sizes, such when a company deploys LLMs in production since they usually do not have memory problems because LLMs are distributed across 8 GPUs.
>
> **Q6: Can you compare the performance between the quantized GEMM with 4-bit SpQR and the quantized GEMM with a normal 4-bit weight quantization method? [...]**
>
> **A6**: The currently best methods for 4-bit quantization are about 3.5x faster than 16-bit methods. SpQR is 1.15x faster than 16-bit methods. As such 4-bit quantization without outliers is up to 3.04x faster than SpQR. However, the performance of SpQR is much better while also compressing the model more strongly, that is close to 3.4 bits per parameter on average. For example, at 3.4 bit the memory savings compared to 4-bit techniques are about 1.3x better and the performance gap in terms of perplexity between 16-bit and 4-bit methods is halved. Taking these factors into consideration, SpQR is pretty close in speed to other methods, because one would need more bits (~5 bits) to achieve similar performance which is about 50% more GPU memory. So for example, SpQR enables to run Falcon 180B on a single GPU, but to get the same performance with regular quantization methods one would need to use two GPUs to fit this model. Since models keep increasing in scale over time, the memory benefit is an important factor.

---

### Official Review · Reviewer_YH3C · 2023-11-07

**Soundness:** 3 good
**Presentation:** 3 good
**Contribution:** 2 fair
**Rating:** 6
**Confidence:** 4

**Summary:**

The paper presents Sparse-Quantized Representation (SpQR), a compression technique for large language models (LLMs) that reduces the weight to 3-4 bits per parameter while maintaining near-lossless accuracy. SpQR isolates and retains outlier weights at high precision and compresses the rest, effectively fitting larger LLMs on devices with limited memory. The approach is shown to preserve model accuracy, reduce memory usage by over 3.4x, and increase inference speed by 15-30% compared to traditional 16-bit models.

======Update 11/26======  Most of my concerns have been addressed through the author's rebuttal and I am revising my score to a 6.

**Strengths:**

- SpQR can compress LLM weights to 3-4 bits per parameter with less than 1% relative accuracy loss.
- It not only reduces the memory footprint by over 3.4x but also speeds up inference by 15-30% compared to 16-bit models.

**Weaknesses:**

- SpQR's methodology, which focuses on outlier management and mixed-precision quantization. Prior works have already delved into the impact of outliers on quantization [\[1\]](https://arxiv.org/pdf/2306.03078.pdf) and the use of mixed-precision quantization [\[2\]](https://openaccess.thecvf.com/content_CVPR_2019/papers/Wang_HAQ_Hardware-Aware_Automated_Quantization_With_Mixed_Precision_CVPR_2019_paper.pdf) to optimize resource use. Hence, SpQR represents an evolutionary improvement in engineering work on LLM weight compression, refining rather than making some novel change in the field.
- The two-step quantization process might be more complex than traditional methods, potentially leading to more involved implementation and tuning.
- While the paper claims efficiency improvements, the actual gains may depend on specific model architectures and deployment scenarios, and there could be overheads associated with managing the sparse representations.
- The benefits of SpQR are maximized with specific hardware capabilities, which may not be universally available.

\[1\] Zhao et al., "Improving Neural Network Quantization without Retraining using Outlier Channel Splitting."

\[2\] Wang et al., "HAQ: Hardware-aware Automated Quantization with Mixed Precision."

**Questions:**

Could you clarify how the 3-bit matrix multiplication was executed on the A100 GPU architecture? Was the focus primarily on optimizing memory access by compressing the weights, while the computation itself was conducted using 4-bit matrix multiplication to balance performance and memory efficiency?

---

> ### Author Response · Authors · 2023-11-18
> **Thank you for your review!**
>
> **Q1: SpQR's methodology, which focuses on outlier management and mixed-precision quantization [...] to optimize resource use. Prior works have already delved into the impact of outliers on quantization [...]**
>
> **A1**: Compared to prior work mentioned we study large language models (LLMs) which have very different outliers patterns than the smaller models common 3 years ago. Please see LLM.int8()[1], for a discussion on how LLM outliers differ from small scale networks. We extend the analysis in [1] to more complex outliers structures. This analysis is novel and goes beyond the structures discovered in earlier work.
>
> **Q2: Hence, SpQR represents an evolutionary improvement [...] rather than making some novel change in the field.**
>
> **A2**: Our analysis is novel and the structures that we discover have not been documented in the literature. Furthermore, the algorithm that we develop is novel. We use a combination of outlier-aware Hessian optimization to find the best quantizations layer-by-layer. Our method is also one of the first, among parallel work, that is able to quantize LLMs to 3-bits without major degradations in performance. All in all, we make major advances in analysis, algorithms, while also providing state-of-the-art results.
>
> **Q3: The two-step quantization process might be more complex than traditional methods, potentially leading to more involved implementation and tuning.**
>
> **A3**: Yes our approach is more involved, but so are comparable approaches. The further we compress LLMs the more complex the approach will be. We believe research on LLM compression works in waves: first complicated approaches reveal new insights, then the research challenge is to simplify these approaches while maintaining performance. The fact that our approach is complex does not diminish the insights and strong results that our work provides. As such, it is a strong basis for any future work on simpler methods.
>
> **Q4: While the paper claims efficiency improvements, the actual gains may depend on specific model architectures and deployment scenarios, and there could be overheads associated with managing the sparse representations.**
>
> **A4**: Since the sparse representations only vary between 0.35% to 1.0% the fraction of outliers does not change much and the performance is almost the same for all architectures. The biggest difference that we see is for Falcon 180B which requires more outliers than other models, about 1%, while Llama v1/v2 require ~0.35-0.45%. This also demonstrates that differences between architectures are very minimal. Please see “Sparse GPU Kernels for Deep Learning”[2) for a discussion on how sparsity level relates to performance of sparse matrix multiplication. Note that for SpQR, our implementation is more effective compared to the implementation presented in [2].
>
> **Q5: The benefits of SpQR are maximized with specific hardware capabilities, which may not be universally available.**
>
> **A5**: SpQR supports has similar performance on all GPUs from the past 10 years. This is so, because the main bottleneck during inference is the memory bandwidth of the GPU rather than other factors (such as FLOPS performance of the GPU). This means that for GPUs, the hardware capabilities are not important. However, we agree that SpQR would be very slow for some hardware, for example TPUs, but since GPUs are widespread we think it is fair to have an implementation which is only fast on GPUs but not TPUs.
>
>
> **Q6: Could you clarify how the 3-bit matrix multiplication was executed on the A100 GPU architecture?**
>
> **A6**: We pack 3-bit integers into 32-bit aligned float values. Once these values are loaded into registers, we perform the dequantization to 16-bit values. We then perform a matrix multiplication with 16-bit inputs and 16-bit weights in registers. The sparse portion of the matrix multiplication is currently done separately. This means we add the results of the sparse matrix multiplication on top of the regular quantized matrix multiplication.
>
> **Q7: Was the focus primarily on optimizing memory access by compressing the weights [...]?**
>
> **A7**: Yes, in this work we do not try to optimize mean FLOPS utilization but we want to try minimize the memory footprint of the weights. This is most useful if the model is deployed for a single user, for example, on a personal computer. In this case, the batch size is 1 since the LLM is queries with one request per time. In this case, the majority of the latency is because the GPU/processor needs to load the weights from GPU memory into registers. We can accelerate this by compressing the weights. In this case, compressing the activations will have no benefit in terms of runtime speed. Activation compression is only beneficial if we use a large batch size (>32) and we use 4-bit or 8-bit tensor cores.
>
> [1] Dettmers et al., LLM.int8(): 8-bit Matrix Multiplication for Transformers at Scale
>
> [2] Gale et al., Sparse GPU Kernels for Deep Learning

---

### Author Response · Authors · 2023-11-21
**Discussion Reminder**

Dear Reviewers,

As the discussion period is drawing to a close and we are yet to receive any reviewer comments, we would like to post a gentle reminder regarding our response. Specifically, we would be happy to receive your feedback regarding the series of additional results we posted (e.g., results on GSM and MMLU and comparisons with GPTQ variants), as well as the individual responses to each of the reviewer questions.

Best regards,\
The SpQR authors

---

### Comment · Area_Chair_w4oQ · 2023-11-22
**Less than one day**

Dear Reviewers,

If you have already responded to authors rebuttal, Thank you!
If not, please take some time, read their responses and acknowledge by replying to the comment. Please also update your score, if applicable.

Thanks everyone for a fruitful, constructive, and respectful review process.

Cheers, Your AC!

---

### Meta-Review · Area_Chair_w4oQ · 2023-12-10

**Metareview:**

This paper proposes a new quantization method that compresses LLM weights to 3-4 bits per parameter with minimal accuracy loss. It works by isolating outlier weights and storing them in higher precision, but, compressing all other weights, allowing over 3.4x compression and 15-30% faster inference compared to the baseline 16-bit models. Experiments on models up to 65B parameters demonstrate near-lossless perplexity and the ability to run large models on commodity GPUs.

Strengths:
- A nice and comprehensive analysis into outlier weight patterns of LLMs.
- Achieves a very good compression to 3-4 bits per parameter with tiny perplexity losses below 1%.
- Demonstrating real implemented speedup rather than only theoretical speedup

Weaknesses:
- The additional cost may not be trivial, a relatively complex 2-stage quantization and sparse matrix multiplication may increase implementation difficulty.
- Unstructured sparsity pattern less hardware friendly compared to dense formats. Much of the benefits rely on specific GPU capabilities not universally available.
- Less solid evaluations and baseline and experimental setup

To improve the paper authors, may want to simplify the proposed quantization method or approximate it to ease adoption for general case. To increase the impact, optimizing sparsity patterns for broader hardware efficiency and demonstrate gains across more hardware configurations is a good next step.

**Justification For Why Not Higher Score:**

Two main reasons prohibits higher score: 1) Less clarity on the effectiveness and general applicability of the method due to complex implementation and hardware dependence. 2) Less clarity on the experimental setup and week baselines and evaluation protocols.

**Justification For Why Not Lower Score:**

The paper has insights to offer that may pave the way for future research in this direction.

---

### Decision · Program_Chairs · 2024-01-16

Accept (poster)